# Flow Matching-Based Autonomous Driving Planning with Advanced Interactive Behavior Modeling

**Tianyi Tan**[1]* **Yinan Zheng**[1]*† **Ruiming Liang**[2]§ **Zexu Wang**[1]
**Kexin Zheng**[3]§ **Jinliang Zheng**[1] **Jianxiong Li**[1] **Xianyuan Zhan**[1]‡ **Jingjing Liu**[1]‡

[1] Institute for AI Industry Research (AIR), Tsinghua University
[2] Institute of Automation, Chinese Academy of Sciences
[3] The Chinese University of Hong Kong
{tanty22,zhengyn23}@mails.tsinghua.edu.cn
zhanxianyuan@air.tsinghua.edu.cn

## Abstract

Modeling interactive driving behaviors in complex scenarios remains a fundamental challenge for autonomous driving planning. Learning-based approaches attempt to address this challenge with advanced generative models, removing the dependency on over-engineered architectures for representation fusion. However, brute-force implementation by simply stacking transformer blocks lacks a dedicated mechanism for modeling interactive behaviors that are common in real driving scenarios. The scarcity of interactive driving data further exacerbates this problem, leaving conventional imitation learning methods ill-equipped to capture high-value interactive behaviors. We propose *Flow Planner*, which tackles these problems through coordinated innovations in data modeling, model architecture, and learning scheme. Specifically, we first introduce fine-grained trajectory tokenization, which decomposes the trajectory into overlapping segments to decrease the complexity of whole trajectory modeling. With a sophisticatedly designed architecture, we achieve efficient temporal and spatial fusion of planning and scene information, to better capture interactive behaviors. In addition, the framework incorporates flow matching with classifier-free guidance for multi-modal behavior generation, which dynamically reweights agent interactions during inference to maintain coherent response strategies, providing a critical boost for interactive scenario understanding. Experimental results on the large-scale nuPlan dataset and challenging interactive interPlan dataset demonstrate that *Flow Planner* achieves state-of-the-art performance among learning-based approaches while effectively modeling interactive behaviors in complex driving scenarios. Official implementation can be found in https://github.com/DiffusionAD/Flow-Planner.

## 1 Introduction

Ensuring safe and reliable planning remains the highest priority for autonomous driving systems in real-world deployment [47]. However, exceptional challenges stem from complex interactions among traffic participants exhibiting multi-modal driving behaviors [15, 42], with the difficulty compounding as the number of participants increases. While conventional rule-based approaches [16, 50] can

---

*Equal Contribution.
†Project Lead.
‡Corresponding Author.
§Work done during internships at Institute for AI Industry Research (AIR), Tsinghua University.

effectively handle most driving scenarios through explicit human-defined constraints and numerical optimization, they are constrained by fundamental limitations, often demanding substantial human engineering efforts and exhibiting poor generalization capability in highly dynamic environments [18]. In contrast, learning-based methods aim to directly learn expert strategies for handling highly interactive scenarios from real-world driving data [3, 19]. These methods have emerged as the dominant choice in both academia [24, 57] and industry [1, 27], with the expectation that they can achieve reasonable planning through increased data volume and model parameters [5].

To model sophisticated interactive behaviors in complex traffic scenarios, a learning-based planner must generate trajectories that simultaneously address immediate interactions, anticipate future behaviors of critical traffic participants, and maintain temporally consistent kinematics [19]. This process critically depends on effective temporal and spatial fusion with scene information. However, the heterogeneity of different elements, particularly static map information and dynamic agents' histories, imposes stringent requirements on the fusion mechanism. Early approaches [25, 39] relied on human priors and over-engineered architectures to capture interactions among traffic participants, but these methods proved difficult to scale and often delivered suboptimal performance. While recent work has adopted transformer-based architectures [9] and generative models [57, 35] to improve scalability and performance, vanilla transformer implementations often fail to effectively capture intricate interdependencies among heterogeneous information [14, 54]. In complex scenarios, extensive redundant information can obscure critical traffic participant information during fusion [57], primarily because these architectures lack specialized designs for interaction modeling.

Moreover, the scarcity of high-quality interactive scenarios in training data leads to a critical limitation: naive behavior cloning methods may converge to biased distributions that fail to capture interactive driving behaviors [26], frequently resulting in safety-critical failures during closed-loop evaluation [13]. Although auxiliary losses can help penalize undesirable behaviors, as commonly done in imitation learning [1, 8, 29, 6], they typically compromise training stability and require careful, case-by-case design. Besides, some methods incorporate prior knowledge, such as pre-searched reference line [8], anchor trajectory [6, 36] and goal point [53], to guide more diverse driving behaviors, but often fail to account for legitimate interactive behaviors that conflict with these structural priors. Alternatively, reinforcement learning [32, 4] can autonomously learn interactive behaviors through trial-and-error, but introduces new challenges including meticulous reward engineering [31, 33] and safety assurance during exploration [56].

To address these challenges, we propose *Flow Planner*, an advanced learning-based framework melding coordinated innovations in data modeling, architecture design, and learning schemes to enhance interactive driving behavior modeling for autonomous driving planning. Specifically, we first break down the complexity of full-trajectory modeling by decomposing it into overlapping segments. This fine-grained tokenization preserves kinematic continuity within each segment while enabling localized feature extraction through segment-specific token representations. Second, our framework enhances interactive behavior modeling through spatiotemporal fusion of scene and planning tokens. Inspired by scale-adaptive attention [38], it dynamically optimizes receptive fields for each token to enable effective extraction of critical information. This process also projects heterogeneous conditions from different scene inputs into a unified representation space, further improving performance. Finally, to enable multi-modal behavior generation in complex driving scenarios, we adopt flow matching loss [37], which offers simpler implementation and faster convergence compared to diffusion-based approaches [57]. Building upon classifier-free guidance [22, 37], our framework dynamically reweights neighboring agent interactions during inference to maintain coherent planning strategies. Compared to naive behavior cloning approaches, this scene-information-enhanced generation mechanism provides substantial improvements in interactive scenario understanding. Experimental results on the large-scale real-world benchmark nuPlan and challenging interactive benchmark interPlan show that *Flow Planner* establishes state-of-the-art performance in closed-loop evaluation among learning-based planners, while demonstrating human-like interactive behaviors in complex traffic scenarios.

## 2 Related Work

**Rule-based Planning Methods**. Rule-based planners determine driving behaviors through manually specified rules, offering interpretable and practically viable solutions that have been validated in both simulation [13] and real-world deployments [16, 34, 51]. Despite their widespread use, these

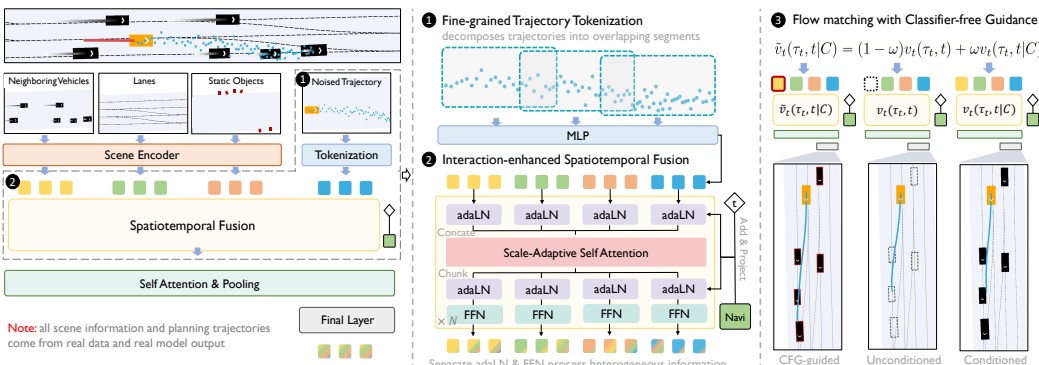

Figure 1: Overview of the *Flow Planner* framework.

methods rely heavily on handcrafted heuristics, which limits their scalability and effectiveness in complex interactive driving scenarios.

**Learning-based Planning Methods**. Conventional imitation learning-based methods for autonomous driving initially relied on CNN [2, 32, 21] or RNN [1] architectures. While modern transformer-based approaches [24, 27, 29, 6] offer improved scalability, they introduce new challenges in interactive behavior modeling. Specifically, effective interactive behavior modeling requires an efficient fusion mechanism to handle heterogeneous scene information, where spatial-temporal interactions between different scene elements (e.g., lanes and neighboring agents) cannot be adequately resolved through simple parameter stacking [9, 46, 41]. While some approaches employ complex architectural designs [25, 39], these solutions often limit the transformer's scalability potential. Although auxiliary loss functions [1, 8, 29, 6] can encourage safe behaviors in complex scenarios through penalties, they fundamentally fail to develop true interactive driving capabilities in models and may result in overly conservative behaviors [56]. Reinforcement learning [31, 33] provides a solution to achieve interactive behavior beyond data capabilities, but due to limitations such as safety exploration and reward design, the application of autonomous driving still remains at the simulation level [12, 4] or simple real-world scenario [32, 44]. Thus, how to improve the model's ability to capture interactive driving behavior and further improve the performance of imitation learning remains a challenge.

**Generative Models for Planning**. Recent work has demonstrated generative models' strong expressiveness for autonomous driving tasks [57, 30, 58]. Autoregressive approaches [46, 7], as a class of generative modeling methods, enable each timestep's action to incorporate both environmental information and previous actions, thereby enhancing modeling capacity. But these methods inherently suffer from error accumulation over time. Diffusion models [45, 23] and flow matching models [37], as the most advanced generative models, share similar denoising processes for generating data from noise. Both demonstrate strong expressiveness in modeling complex distributions [10], making them particularly suitable for autonomous driving behavior modeling. However, existing methods still rely heavily on prior knowledge (e.g., anchor trajectories [36] or goal points [53]). Although such strong priors can stimulate multi-modal driving behaviors, they tend to cause the model to neglect interactive behaviors with traffic participants. *Diffusion Planner* [57] attempts to improve interactive behavior modeling by jointly generating both the ego planning and predictions for neighboring vehicles without requiring prior knowledge. However, its interactions are constrained to a fixed number of nearest vehicles, which restricts potential interactions with important distant agents, and the architecture lacks specific designs for an effective fusion mechanism. Therefore, there still lacks a design to further push the limits of generative models for interactive behavior modeling.

## 3 Method

In this section, we provide a brief introduction to trajectory generation in planning tasks and present *Flow Planner*, a novel approach that emphasizes interactive behavior modeling, as shown in Figure 1. From the data modeling perspective, we propose fine-grained trajectory tokenization to achieve expressive trajectory modeling. Subsequently, we design a well-curated architecture that enhances interactive behavior modeling through thorough spatiotemporal fusion. Finally, we adopt flow matching with classifier-free guidance to further enhance multi-modal and interactive driving behaviors.

## 3.1 Problem Formulation

Our work primarily focuses on the planning module of autonomous driving systems, utilizing processed perception as input and evaluating planning capabilities through closed-loop testing [19]. The trajectory generation task can be formulated as a conditional generation problem, where the transformation from a source distribution $p(\tau_0)$ (typically standard Gaussian) to a target data distribution $q(\tau_1|C)$ (e.g., planning trajectories $\tau_1$ with scene information $C$) is represented by a probability path. The model $v_\theta(\cdot)$ is trained to predict the velocity along the path connecting sample pairs $(\tau_0, \tau_1)$ drawn from the source and target distributions, using the objective [37]:

$$\mathcal{L} = \mathbb{E}_{t\sim\mathbb{U}(0,1),p(\tau_0),q(\tau_1|C)}||v_\theta(\tau_t,t|C) - v_t(\tau_t,t)||^2, \tag{1}$$

where $\tau_t = \alpha_t\tau_1 + \sigma_t\tau_0$ and the ground truth velocity $v_t(\tau_t,t) = \dot{\tau}_t = \dot{\alpha}_t\tau_1 + \dot{\sigma}_t\tau_0$ is the time derivative of the interpolation between source and target points. The distinction between ODE-based diffusion models and flow matching models lies in their respective designs of probability paths.

## 3.2 Model Architecture

*Flow Planner* is a transformer-based generative model for motion planning that effectively fuses noisy trajectory $\tau_t$ with scene condition $C$. The architecture overview is presented in Figure 1.

**Scene Encoder**. We consider the vectorized driving scene information. For each neighboring agent (e.g., vehicle, pedestrian, cyclist), we represent its past $T$ timestep information as $F_{\text{neighbor}} \in \mathbb{R}^{T \times H_{\text{neighbor}}}$, where $H_{\text{neighbor}}$ is the dimension of state information including coordination, velocity, size, and the type of different agents. For each lane, we interpolate the centerline into $N$ uniformly distributed points. Each point contains coordinates, corresponding boundary vectors, speed limit information, and traffic light status, represented as $F_{\text{lane}} \in \mathbb{R}^{N \times H_{\text{lane}}}$. To maximize scenario information preservation while maintaining model efficiency, we employ separate MLP-Mixer architectures [48] to encode both agent and lane features, following [57]:

$$F = F + \text{MLP}_{seq}(F), F = F + \text{MLP}_{feat}(F), \tag{2}$$

where $\text{MLP}_{seq}$ and $\text{MLP}_{feat}$ are two separate MLPs operating on sequence and feature dimensions respectively, and $F$ is the information of neighboring agents $F_{\text{neighbor}}$ or lanes $F_{\text{lane}}$. Additionally, we consider static objects, encoded using another MLP. The navigation information is represented similarly to lane information and is processed by a separate MLP-Mixer.

**Fine-grained Trajectory Tokenization**. We start by rethinking the trajectory tokenization in autonomous driving planning, an area that has received inadequate research attention despite its critical importance. Prevalent methods [57, 9, 36] use a single token to represent the whole trajectory, which maintains kinematic consistency [28] but suffers from inefficient scene context fusion due to over-compression. Alternative approaches utilizing either discrete timestep tokens [11] or autoregressive generation [46] successfully achieve temporal fusion within trajectories. However, these methods inevitably encounter compounding error accumulation, presenting a fundamental limitation for closed-loop autonomous driving applications.

To address these limitations, we propose a balanced modeling framework featuring fine-grained trajectory tokenization that preserves temporal interaction patterns while ensuring consistent behavior across all planning horizons. Specifically, the noised trajectory $\tau_t = (x_1, x_2, ..., x_L)$ introduced in Section 3.1, which comprises $L$ points in total, is first divided into $K$ segments, each containing $L_{seg}$ points. Additionally, the neighboring segments share an overlap with a length $L_{overlap}$ to ensure the consistency and smoothness of the resembled trajectory. Next, a shared MLP is used to transform the noised segments into the ego-trajectory tokens:

$$F_{ego}^k = \text{MLP}\left((x_{l^k}, x_{l^k+1}, ..., x_{r^k})\right), k = 1, 2, ..., K, \tag{3}$$

where $l^k = (k-1)(L_{seg} - L_{overlap})$ is the start timestep of the trajectory segment, and $r^k = (k-1)(L_{seg} - L_{overlap}) + L_{seg}$ is the end timestep. Note that timesteps in trajectories differ conceptually from noise step $t$ in generative models. After that, we adopt the sinusoidal position encoding [52] to inject temporal information. Finally, we concatenate all the segment tokens along the sequence dimension to obtain the ego feature: $F_{ego} = \text{Concat}\left(F_{ego}^1, ..., F_{ego}^K\right)$.

**Interaction-enhanced Spatiotemporal Fusion**. Although fine-grained trajectory tokenization provides more expressive representations for behavior modeling, the efficient fusion of spatiotemporal

information remains unresolved. This process requires bidirectional interaction among numerous information-sparse tokens to enable comprehensive scene understanding and behavior modeling. Another critical challenge lies in effectively fusing these heterogeneous modalities, such as the static information of lanes and the dynamic information of agents. These challenges are what vanilla transformer attention [9, 8, 57] fails to accomplish effectively.

Inspired by recent work in text-to-image generation [14, 54], where cross-modality feature fusion is performed in a unified space, we first process heterogeneous features through separate adaptive LayerNorm (adaLN) modules [43], projecting them into a shared latent space where both timestep conditions and navigation information are injected via modulation mechanisms [57]. Then the processed tokens from one scenario, including the features of lanes $F_{lane}$, neighboring agents $F_{neighbor}$ and ego planning trajectory $F_{ego}$, are concatenated along the sequence dimension:

$$F_{global} = \text{Concat}\left(\text{adaLN}\left(F_{lane}\right), \text{adaLN}\left(F_{neighbor}\right), \text{adaLN}\left(F_{ego}\right)\right). \tag{4}$$

For the subsequent fusion of all tokens $F_{global}$, a critical observation reveals that interaction intensity between traffic participants correlates strongly with their spatial distances. For instance, excessively distant roads and vehicles may introduce noise that degrades planning performance [57]. Inspired by scale-adaptive self-attention [38], we employ learnable receptive fields for individual tokens to incorporate spatial guidance during feature fusion. This is achieved through spatial distance-scaled attention score adjustment:

$$F_{global} = \text{Softmax}(\frac{F_{global}W^Q \left(F_{global}W^K\right)^T}{\sqrt{d}} - \lambda \cdot D)F_{global}W^V, \tag{5}$$

where $W^Q, W^K, W^V$ are the pre-projection weights used to generate query, key, and value for the attention mechanism [52], $D$ is the pairwise Euclidean distance matrix of the tokens, and $\lambda$ is the receptive scaler generated by a simple linear projection of the tokens. Intuitively, tokens representing traffic participants beyond a certain distance are assigned smaller attention scores, making them less influential in the computation and enabling more adaptive resource allocation. Then, the global feature is decomposed into modality-specific tokens, where each modality undergoes distinct adaLN and FFN projections to further mitigate the heterogeneous modality gap:

$$\begin{aligned} F_{lane}, F_{neighbor}, F_{ego} &= \text{Chunk}\left(F_{global}\right), \\ F_{lane} = \Psi\left(F_{lane}\right), F_{neighbor} &= \Psi\left(F_{neighbor}\right), F_{ego} = \Psi\left(F_{ego}\right), \end{aligned} \tag{6}$$

where $\Psi$ is the abbreviation of the FFN(adaLN($\cdot$)). Building upon the foundation established in Eq. (4)-(6), our architecture employs stacked transformer blocks to progressively refine spatiotemporal interactions. Finally, a standard self-attention layer is used for final aggregation. The features are then processed through token pooling to obtain ego planning tokens, followed by a final layer $\Psi$ [43, 57] that transforms these representations into the ego vehicle's planned trajectory.

### 3.3 Guided Trajectory Generation via Flow Matching

With the enhanced model architecture established, we want to push the upper-bound performance for imitation learning in trajectory generation. Compared to conventional behavior cloning, generative models show better expressiveness in modeling multi-modal and interactive driving behaviors [57]. The most compelling aspect of generative models lies in their ability to dynamically reweight conditional signals during inference through classifier-free guidance [22], thereby amplifying conditional influence to achieve superior conditioned generation. Specifically, we enhance the conventional behavior cloning approach by combining the scene-conditioned planning trajectory distribution $q(\tau_1|C)$ with an unconditioned distribution $q(\tau_1)$, yielding an enhanced distribution $\tilde{q}(\tau_1|C) \propto q(\tau_1)^{1-\omega} q(\tau_1|C)^\omega$, where $\omega > 1$ is the weighting parameter that controls the condition signal strength. Intuitively, the model learns both the planning behavior without scene conditions and the behavior with scene conditions, enabling it to implicitly capture the behaviors induced by scene conditions. This approach strengthens the understanding of complex driving scenarios and improves interactive behavior modeling. Building upon this framework, we can sample from distribution $\tilde{q}(\tau_1|C)$ using the guided velocity field as follows [22]:

$$\tilde{v}_t(\tau_t, t|C) = (1 - \omega)v_t(\tau_t, t) + \omega v_t(\tau_t, t|C). \tag{7}$$

The remaining challenge involves training the model to estimate velocities in Eq. (7). We address this by training a single model with Bernoulli-masked conditions as follows [55]:

$$\mathcal{L}_{flow} = \mathbb{E}_{t \sim \mathbb{U}(0,1), b \sim \mathcal{B}, p(\tau_0), q(\tau_1|C)}||\tau_\theta\left(\tau_t, t|(1-b) \cdot C + b \cdot \varnothing\right) - \tau_1||^2. \tag{8}$$

In Eq. (8), we employ the reparameterization trick to directly predict ground-truth trajectories. This formulation proves mathematically equivalent to the velocity prediction loss in Eq. (1) [37]. Additionally, we employ the optimal transport path [40, 49] widely used in flow matching, where the velocity estimation follows $v_\theta(\tau_t, t|C) = (\tau_\theta(\tau_t, t|C) - \tau_0)/t$. The same approach applies to unconditioned velocities $v_\theta(\tau_t, t)$. Finally, we obtain the guided velocity from Eq. (7) and employ an ODE solver for trajectory generation.

Notably, the conditioned information can be flexibly selected in our framework. In practice, we specifically mask neighboring vehicle information in Eq. (8), as we empirically find this to be most critical for interactive driving behavior. As shown in Figure 1, the unconditioned model fails to account for neighboring vehicles, while the conditioned model overreacts to the nearest vehicle with overly conservative planning behavior. In contrast, our classifier-free guided approach successfully generates reliable planning results that appropriately respond to neighboring vehicle dynamics.

### 3.4 Implementation Details

In our implementation, the entire scenario is first transformed into an ego-centric front-left-up coordinate. To ensure training stability and input consistency, input data as well as ground truth trajectories are first normalized using static measures extracted from training data. In addition, data augmentation [57] has been demonstrated as effective in improving the robustness of planning outputs. During training, the ego vehicle's state at the current frame is first randomly perturbed, and a quintic polynomial is used for interpolating a new trajectory, serving as the ground truth of the augmented sample. During inference, the second-order midpoint method [37] is used for the flow ODE solver. In addition, to ensure the consistency of the generated trajectory, we introduce an extra consistency loss $\mathcal{L}_{consist}$ on the segment overlap during training. Specifically, given the predicted segments, we apply L2 loss on the overlap of neighboring predicted segments:

$$\mathcal{L}_{consist} = \frac{1}{K-1} \sum_{k=1}^{K-1} \|\hat{\tau}^{k:k+1} - \hat{\tau}^{k+1:k}]\|^2, \tag{9}$$

where $\hat{\tau}^{k:k+1}$ denotes the portion of the predicted trajectory from segment $k$ that overlaps with segment $k+1$, and $\hat{\tau}^{k+1:k}$ represents the portion of the predicted trajectory from segment $k+1$ that overlaps with segment $k$. This consistency loss is added to the flow matching loss with a loss scaling hyperparameter $\alpha > 0$ to form the final objective $\mathcal{L} = \mathcal{L}_{flow} + \alpha \cdot \mathcal{L}_{consist}$. Note that the converged model is equivalent to the one supervised by merely the vanilla flow matching objective. During inference, the predicted values of the overlapping areas are averaged in a straightforward manner to generate the final prediction.

## 4 Experiments

In this section we conducted extensive experiments to demonstrate the performance of our model. We will start by briefly introducing the benchmarks on which the experiments are performed, and the primary baselines we compared our model with. Next, we will focus on several representative cases that intuitively illustrate the advantage of our method, comparing it with the previous state-of-the-art method in specific scenarios. Then, we will delve into the details of our designs and showcase their effectiveness through a further ablation study.

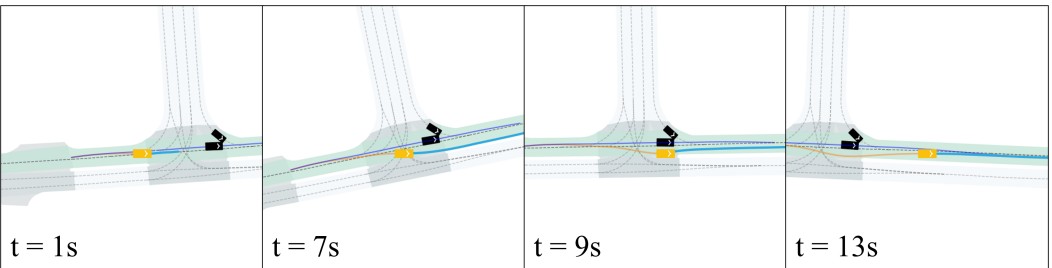

Figure 2: A typical out-of-distribution scenario in interPlan benchmark: *nudging around crashed vehicles*. *Flow Planner* demonstrate strong scene understanding ability and generation adaptability in the situation that is totaly unseen in the training data.

## 4.1 Experimental Setup

**Benchmarks**. In this study, our model is trained on nuPlan [19] featuring a large-scale real-world driving dataset collected across four different cities. This dataset covers up to 75 scenario types, including diverse scenarios with multi-vehicle interaction and complex lane structures. Specifically, the model is trained on the 1M training split, following [57]. For evaluation, the model is tested on both nuPlan and interPlan [20], and we mainly focus on the closed-loop performance of the planners, where an LQR controller is used for simulation. We test the model in both non-reactive and reactive settings using the following three benchmarks in nuPlan: (1) Val14 [13], a validation dataset with 1118 scenarios in total; (2) Test14-random [9]: over 200 randomly selected scenarios from the scenario types assigned by the nuPlan Planning Challenge; and (3) Test14-hard [9]: a collection of the worst-performing scenarios by rule-based PDM [13], comprising 272 scenarios. Each of the three benchmarks covers 14 different types of scenarios respectively. We argue that the Val14 benchmark can evaluate model's performance and capability more comprehensively owing to its sufficient capacity, while Test14-random benchmark may introduce extra uncertainty, hindering the fairness of evaluation. Whereas *Flow Planner* achieves overall state-of-the-art performance across three benchmarks, we primarily leveraged Val14 for further experiments. In addition, we evaluate our model using the full-scale interPlan benchmark, which contains 335 challenging interactive scenarios with specially augmented traffic agents quantity and behavior, as shown in Fig. 2, in reactive mode. This benchmark highlights model's capability of modeling interactive behavior in complex and unexpected scenarios, providing a more appropriate evaluation of our method.

**Baselines**. For a comprehensive analysis of the effectiveness of our method, we conduct comparative experiments with prevailing methods, including rule-based, imitation learning, and hybrid methods that refine the learning-based model with rule-based trajectory post-processing. Specifically, we compare our method with the following baselines:

- *IDM* [50]:a classic rule-based method, also used for neighboring vehicles control in closed-loop reactive evaluation;
- *PDM* [13]:the first place of nuPlan contest, which proposed a rule-based model (*PDM-Closed*), a learning-based model (*PDM-Open*) as well as a hybrid version (*PDM-Hybrid*), all relying on road centerline;
- *PLUTO* [8]: an imitation learning-based model with extra contrastive objectives and post-processing;
- *GameFormer* [25]: a transformer-based model for interactive prediction based on game-theory, with extra refinement process;
- *Diffusion Planner* [57]: the state-of-the-art model based on diffusion model for multi-modal trajectory generation.

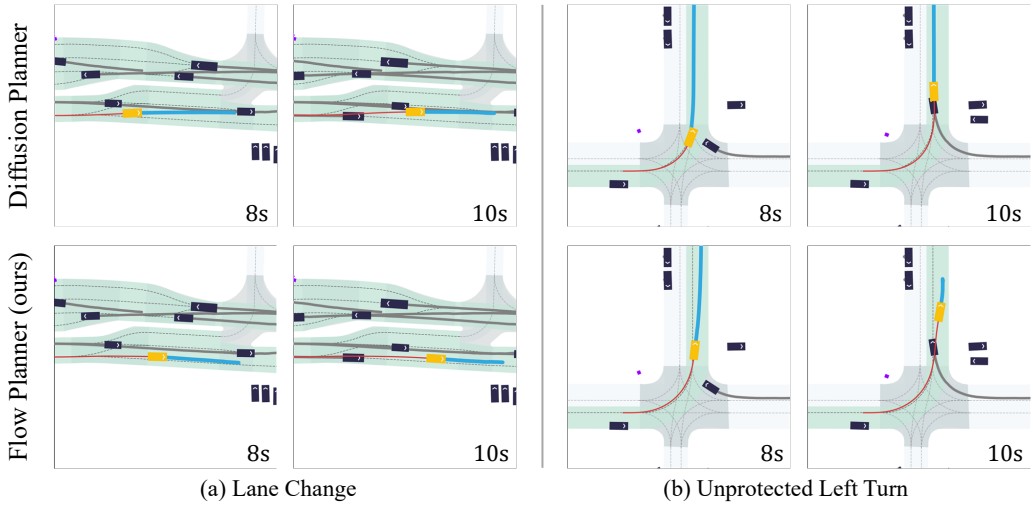

Figure 3: Visualization of interaction behaviors. Two challenging scenarios with distinctive interactions in closed-loop testing, including: (a) *changing lane* and (b) *unprotected left turn* in the closed-loop test. The trajectories illustrated here include: the future planning of ego vehicle, the ego history, and the neighbor history.

Table 1: The overall performance of *Flow Planner* and other learning-based baseline models. Evaluation is conducted in both non-reactive and reactive mode. The highest score of each benchmark is marked with ▢, and the second-best scores appear in **bold**. In addition, we use * to mark the methods directly leveraging road centerline extracted from the map.

| Type | Planner | Val14 | | Test14-hard | | Test14 | |
|------|---------|-------|-----|-------------|-----|--------|-----|
| | | NR | R | NR | R | NR | R |
| Expert | Log-replay | 93.53 | 80.32 | 85.96 | 68.80 | 94.03 | 75.86 |
| Rule-based & Hybrid | IDM | 75.60 | 77.33 | 56.15 | 62.26 | 70.39 | 74.42 |
| | PDM-Closed | 92.84 | 92.12 | 65.08 | 75.19 | 90.05 | 91.63 |
| | PDM-Hybrid | 92.77 | 92.11 | 65.99 | 76.07 | 90.10 | 91.28 |
| | GameFormer | 79.94 | 79.78 | 68.70 | 67.05 | 83.88 | 82.05 |
| | PLUTO | 92.88 | 76.88 | 80.08 | 76.88 | 92.23 | 90.29 |
| | Diffusion Planner w/ refine. | 94.26 | 92.90 | 78.87 | 82.00 | 94.80 | 91.75 |
| | Flow Planner w/ refine. (Ours) | 94.31 | 92.38 | 78.64 | 80.25 | 94.79 | 92.40 |
| Learning-based | PDM-Open* | 53.53 | 54.24 | 33.51 | 35.83 | 52.81 | 57.23 |
| | GameFormer w/o refine. | 13.32 | 8.69 | 7.08 | 6.69 | 11.36 | 9.31 |
| | PlanTF | 84.27 | 76.95 | 69.70 | 61.61 | 85.62 | **79.58** |
| | PLUTO w/o refine.* | 88.89 | 78.11 | 70.03 | 59.74 | 89.90 | 78.62 |
| | Diffusion Planner | **89.87** | **82.80** | **75.99** | **69.22** | **89.19** | 82.93 |
| | Flow Planner (Ours) | 90.43 | 83.31 | 76.47 | 70.42 | 89.88 | 82.93 |

Table 2: Performance of different planners in specific scenarios of Val14, where `"Intersection"` corresponds to the `"Starting straight traffic light intersection traversal"` scenario.

| Planner | Starting right turn | Starting left turn | Intersection | Waiting for pedestrian |
|---------|---------------------|--------------------|--------------|------------------------|
| PLUTO (w/o refinement) | 80.19 | 86.51 | 90.08 | 84.65 |
| Diffusion Planner | 78.10 | 87.96 | 94.57 | 91.65 |
| Flow Planner (Ours) | 83.23 | 90.60 | 94.81 | 93.25 |

## 4.2 Main Results and Case Study

**Main Results**. The overall evaluation results are shown in Table 1. *Flow Planner* achieves state-of-the-art performance in the learning-based settings, without rule-based refinement. It is noteworthy that *Flow Planner* achieves **90.43** on Val14, which is the largest and most representative benchmark of the three. As far as we know, it is the first learning-based method to surpass the 90-score mark without any prior knowledge on this benchmark, while other learning-based models require extra rule-based refinement to reach 90+ performance. With the similar post-processing module in [57], our model *Flow Planner w/ refine.* achieved competitive performance on the nuPlan benchmark compared with previous rule-based and hybrid methods. To find out the specific improvement brought by our method, we further report the performance of *Flow Planner* and baseline methods in several specific types of scenarios in nuPlan. As is shown in Table 2, *Flow Planner* outperforms the two strong baselines significantly in scenarios where interaction with neighboring vehicles is frequent, including unprotected left turns and traffic light intersections. In addition, on the interPlan benchmark in which most scenarios require dense interaction with the environment, as is shown in Table 3, we achieve **8.92**-point improvement over Diffusion Planner, demonstrating the strong capability of interactive modeling. It can also be inferred from the table that *Flow Planner* exhibits robust interactive behavior modeling ability even when interacting with jaywalking pedestrians, whose behavior is tricky to predict due to the lack of relevant data and their volatile movement.

**Case Study**. To further demonstrate the effectiveness of our method, we select representative interactive scenarios to compare the behavior of different models in the closed-loop testing. As is shown in Figure.3, we selected two scenarios from nuPlan benchmark, and report the different behavior of our *Flow Planner* and *Diffusion Planner*. Specifically, in (a), *Diffusion Planner* failed to account for a vehicle approaching quickly from the rear-left. It changed lanes despite being

Table 3: Performance of different planners on interPlan and scores on specific scenarios

| Planner | Overall Score | Nudge Around | High Traffic Density | Jaywalk |
|---|---|---|---|---|
| PlanTF | 47.70 | 49.40 | 58.85 | 33.94 |
| PLUTO (w/o refinement) | 58.47 | 71.56 | 67.25 | 25.48 |
| Diffusion Planner | 52.90 | 60.48 | 49.71 | 26.20 |
| Flow Planner (Ours) | 61.82 | 72.96 | 67.21 | 43.57 |

slower than the approaching car, resulting in a collision. In contrast, *Flow Planner* recognized the fast-approaching vehicle and, realizing that the safety distance was insufficient, aborted the lane change to avoid collision; while in (b), *Diffusion Planner* did not consider a right-turning vehicle from the oncoming lane. It entered the turn at a low speed, leading to a collision. *Flow Planner*, however, identified that the right-turning vehicle would arrive later and chose a higher entry speed, safely completing the turn. More closed-loop planning results where *Flow Planner* exhibits interactive behavior modeling capability are shown in Appendix A.

## 4.3 Ablation Studies

We ablate the key designs of our method to further study their effects on model performance. The overall ablation path is shown in Table 4, where we start with a base model by simply stacking self attention layers to form the decoder while keeping the scene encoder unchanged. The model is then gradually enriched to form the final architecture, so as to better reflect the improvements gained from each component.

Table 4: Ablation path of key components on nuPlan(Val14) and interPlan, and performance on specific interPlan scenarios. Below we use NA for Nudge Around, HTD for High Traffic Density and JW for Jaywalk

| Components | nuPlan(Val14) | interPlan | NA | HTD | JW |
|---|---|---|---|---|---|
| Base | 88.10 | 41.27 | 50.61 | 38.18 | 27.99 |
| + Trajectory Tokenization in Eq. (3) | 88.33 | 44.14 | 43.27 | 48.11 | 52.73 |
| + Scale-Adaptive Attention in Eq. (5) | 88.77 | 46.25 | 51.31 | 44.28 | 34.42 |
| + Separate adaLN & FFN in Eq. (4), (6) | 89.54 | 58.22 | 64.46 | 60.08 | 43.16 |
| + Classifier-free Guidance in Section 3.3 | 90.43 | 61.82 | 72.96 | 67.21 | 43.57 |

Table 5: Ablation on the number of trajectory segments on nuPlan Val14 Benchmark.

| Length | Score |
|---|---|
| 1 | 88.48 |
| 4 | 89.43 |
| 16 | 89.27 |
| 20 | **90.43** |
| 40 | 89.32 |
| 80 | 87.75 |

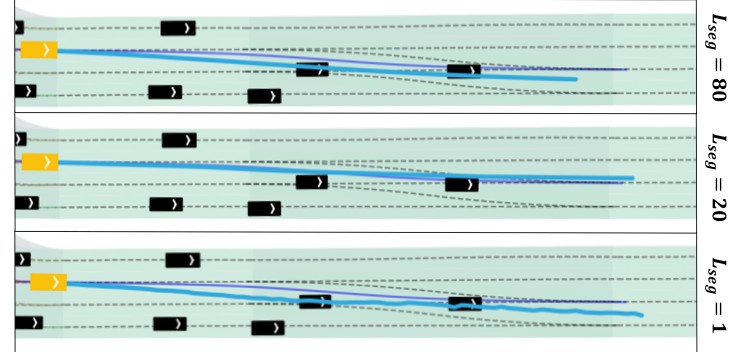

Figure 4: Illustration of the influence of token number on trajectory quality.

**Trajectory Tokenization**. The choice of trajectory segment number and the length of overlap directly affect the balance of consistency and flexibility of the generated trajectory. Therefore, we trained different models with different choices of segment number, and evaluated them on Val14, as shown in Table 5 and Figure 4. Note that the overlap length is set to 0 when the segment length is 1 (fully scattered) or 80 (whole trajectory), and half the length of the trajectory segment in other cases. It can be revealed from the table and figure that as the length of segments increases, the smoothness of the

generated trajectory increases as well as the model performance. However, when the trajectory is coarsely segmented, the trajectory tokens become cumbersome to model the multi-modal distribution of interactive behavior since a single token is responsible for a relatively long horizon of trajectory, which can contain several different interactions with neighboring agents.

**Scale-Adaptive Attention and Separate AdaLN and FFN**. Scale-adaptive attention enables more efficient feature fusion between numerous tokens extracted from the scenario. However, there exists inherent heterogeneity between the features collected from different types of instances in the scenario, and thus the straightforward implementation of adaptive attention does not introduce visible improvement. With the separate adaLN and FFN modules projecting the heterogeneous features into a shared space, a prominent performance improvement can be seen from Table 4 when the scale-adaptive attention is cooperatively implemented.

**Classifier-free Guidance**. The scale of classifier-free guidance is a flexible hyperparameter that can be tuned at inference. Therefore, we ablated different choices of guidance scale using the same checkpoint. The results are illustrated in Table 6. It can be seen from the table that as the scale of guidance increases, the model's performance also improves. However, an overwhelmingly large scale can also lead to deterioration in performance, since the norms of the conditioned and unconditioned velocity are not perfectly aligned, and a scale too large may lead to irreversible deviation on the flow path. Ideally, a proper scale applied during inference can induce even better performance compared with the fully conditioned generation, as is shown in Figure 1.

Table 6: Ablation on the scale of classifier-free guidance on Val14.

| CFG Scale | Score |
|---|---|
| 1.65 | 89.64 |
| 1.70 | 89.89 |
| 1.75 | 90.14 |
| 1.80 | **90.43** |
| 1.85 | 90.00 |
| 1.90 | 89.63 |

## 5    Conclusion

We introduce *Flow Planner*, a novel learning-based framework for autonomous driving planning that advances interactive behavior modeling through three coordinated innovations. First, fine-grained trajectory tokenization enables expressive behavior representation. A well-designed architecture then facilitates comprehensive spatiotemporal fusion to enhance interaction modeling. Finally, flow matching combined with classifier-free guidance captures the multi-modal nature of real driving behaviors, which further attunes to interactive behavior during inference. Over the nuPlan benchmark, *Flow Planner* achieves state-of-the-art closed-loop performance among imitation-learning methods, while demonstrating capability in modeling interactive behaviors in complex driving scenarios. Due to space limit, more discussion on limitations and future direction can be found in Appendix E.

## Acknowledgement

This work is supported by National Key Research and Development Program of China under Grant 2022YFB2502904, and funding from BYD Automotive New Technology Research Institute, Wuxi Research Institute of Applied Technologies, Tsinghua University, under Grant 20242001120, and Xiongan AI Institute.

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

# Appendix

## A    Visualization of Closed-loop Planning Results

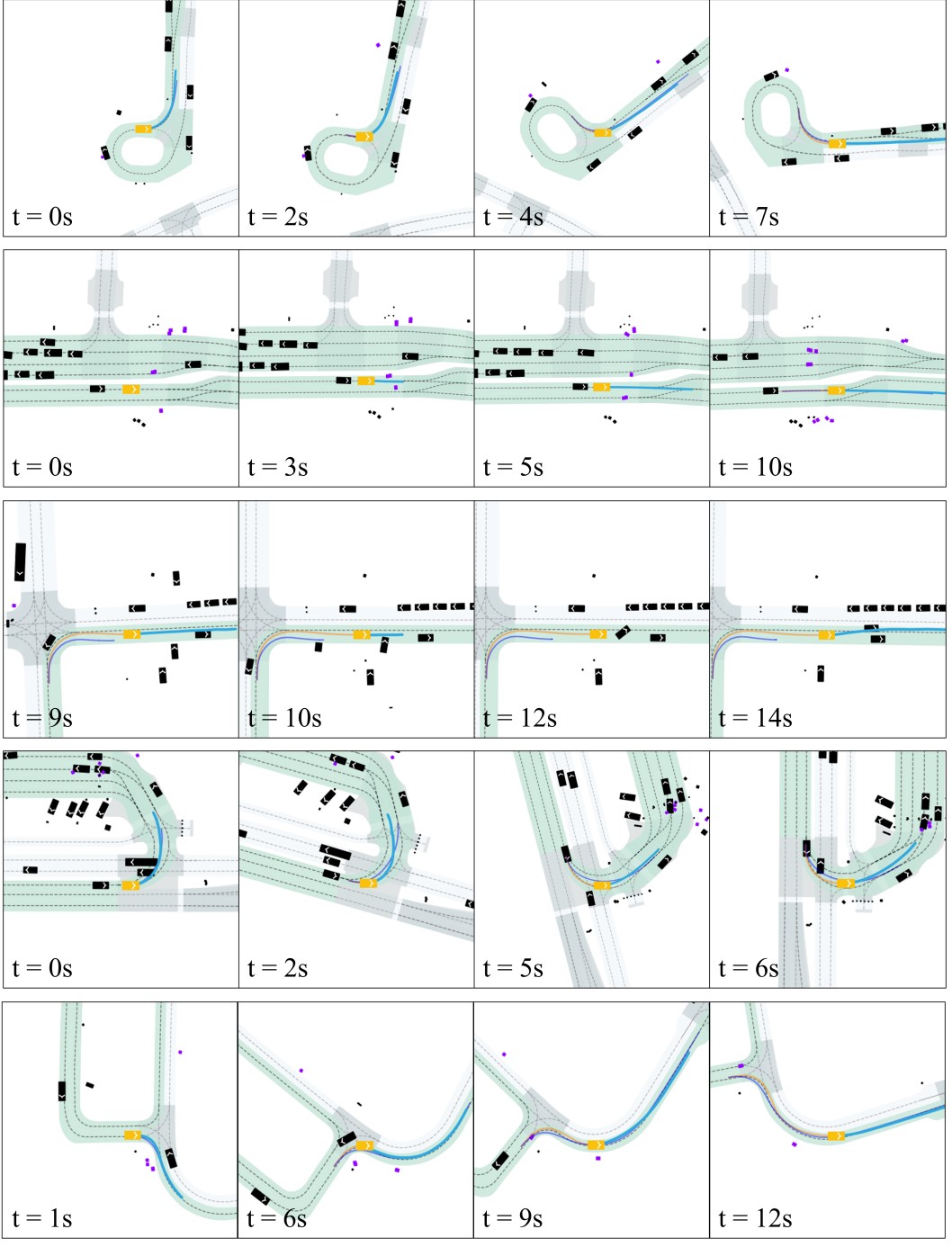

Figure 5: Interaction cases in closed-loop results (part 1).

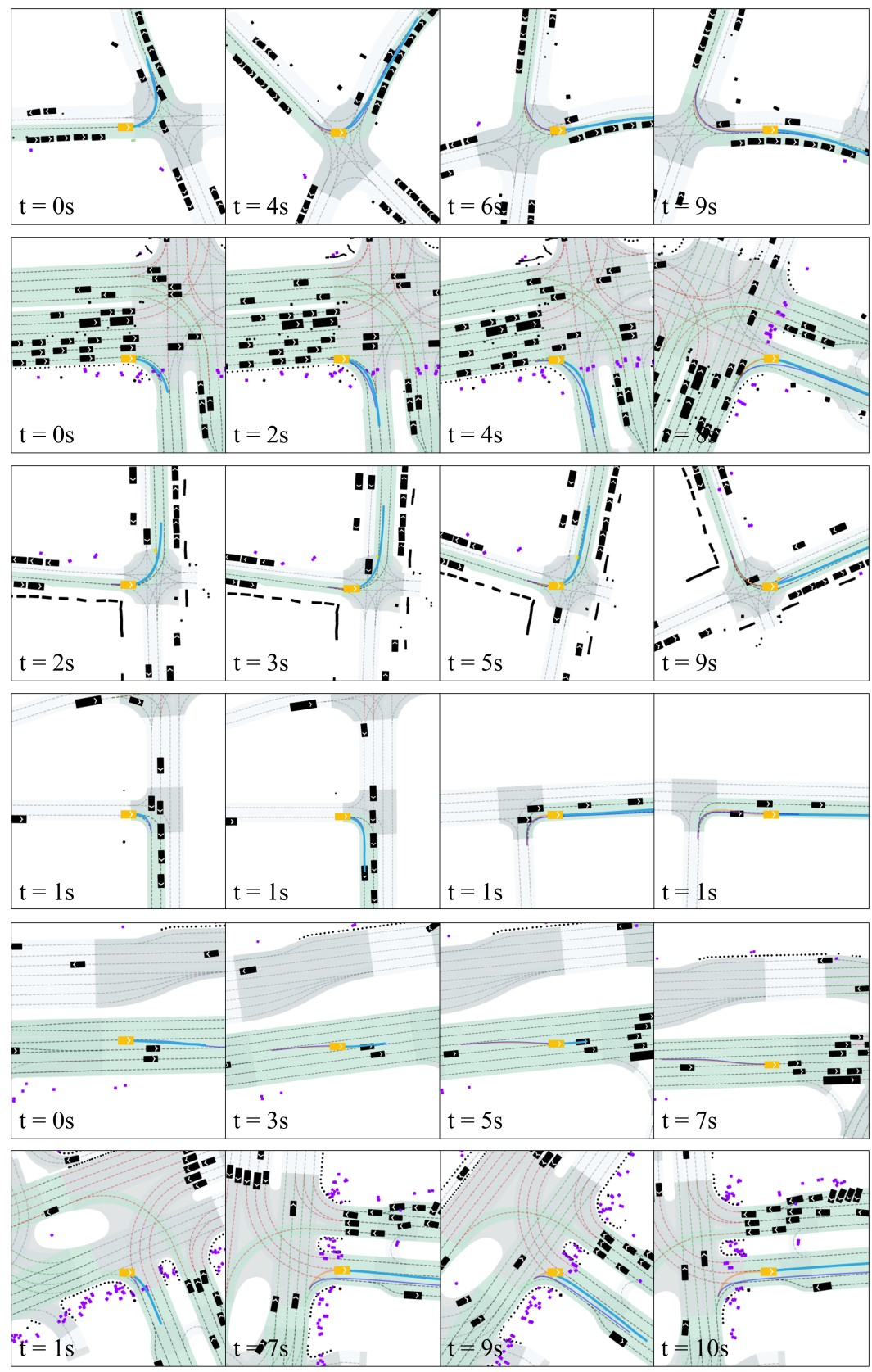

Figure 5: Interaction cases in closed-loop results (part 2).

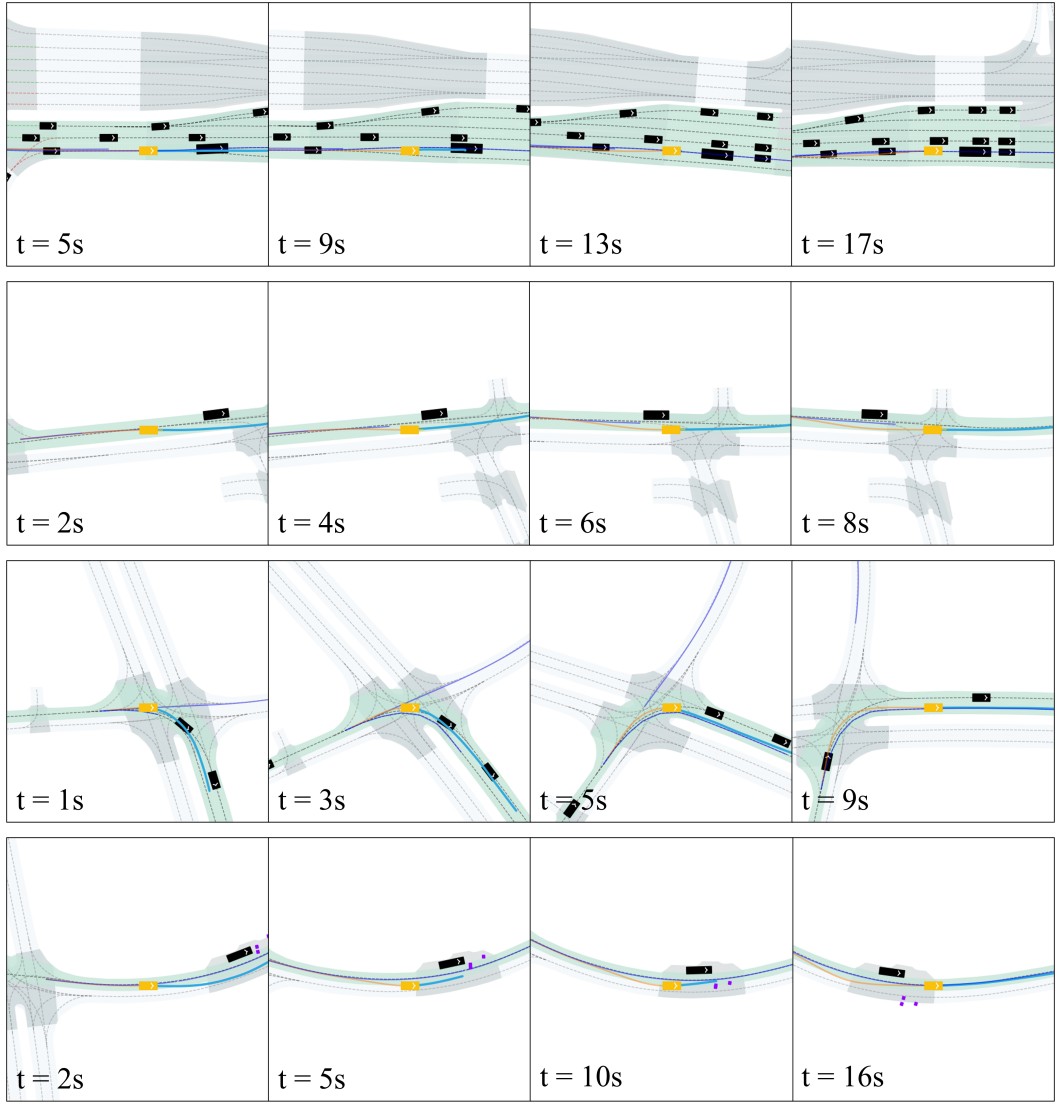

Figure 6: Interaction cases in interPlan closed-loop results.

## B Implementations of Training

**Data Preprocessing**. A scenario in nuPlan contains versatile scene information, whereas not all of it is used for model prediction. In our method, the model takes the lanes, neighboring agents (vehicles, pedestrians, cyclers etc.), navigation information and static objects from the scenario as input. These scenario inputs are collected into tensors once and for all, and can be reused during training.

- *Lanes*: a lane is represented by a sequence of 20 points sampled from the past 2 seconds at 10 Hz, each of which contains the coordinates of the lane center line and lane boundary line, as well as the connecting vector of center line and lane traffic signal; up to 70 lanes are fed into the model;
- *Agents*: the nearest 32 agents to the ego vehicle is encoded as the model input, and each agent contains the historical trajectories in the past 2 seconds at 10Hz, and states in the current frame;
- *Navigation*: represented by 5 lanes along the assigned direction, saved in the same form as lanes;
- *Static Objects*: the static obstacles in the scenario, represented by a vector containing the position in the ego centric coordinates at the current frame and the specific type of the obstacle;

**Transformation and Normalization**. The original data in nuPlan is recorded in the world coordinates, in which the starting point of the ego vehicle can appear in any place, introducing unnecessary

complexities for training. Therefore, to unify the numerical value of model input, we first transform the entire scenario into the ego-centric front-left-up coordinate, in which the state of the ego vehicle at the first frame of each scenario is $(X = 0, Y = 0, \theta = 0)$, and the coordinates of other scenario instances are transformed accordingly. Specifically, given the world coordinates of the ego vehicle $(X, Y, \theta)$, the transformation from the original coordinates $(x, y)$ to the transformed coordinates $(x', y')$ can be formulated as

$$
\begin{bmatrix} x' \\ y' \\ 1 \end{bmatrix} = \begin{bmatrix} \cos\theta & \sin\theta & -X\cos\theta - y\sin\theta \\ -\sin\theta & \cos\theta & X\sin\theta - Y\cos\theta \\ 0 & 0 & 1 \end{bmatrix} \begin{bmatrix} x \\ y \\ 1 \end{bmatrix} \tag{10}
$$

In addition, the original scale of coordinates varies in different scenario. To ensure the numerical stability, we normalized the coordinates and trajectories with statics from training data via z-score along the x-axis and scaled along the y-axis [9, 57]. The model directly predicts the normalized future trajectory, and it is then denormalized into the scale of the real-world trajectory.

**Data Augmentation**. Following previous practice [57], we applied data augmentation for more robust generation. The initial states, including coordinates, velocities and heading angle, of the ego vehicle are first perturbed with random offsets. Then the perturbed initial states and the states at the 20-th frame are used as the boundary condition to solve a new quintic polynomial to replace the first 20 frames of the original ground truth trajectory. Intuitively, this augmentation forces the model to output a feasible future trajectory even when the ego vehicle is not properly placed on the road, enabling self-correction when the vehicle is driving off the road.

## C   Experimental Details

The training is conducted on 8 NVIDIA A6000 GPUs, using the 1M training data split from nuPlan. The model is trained for over 200 epochs with a batch size of 2048. We used AdamW optimizer for training, and the learning rate is set to be $5 \times 10^{-4}$. In addition, we used exponential moving average (EMA) to stablize the training process, with 0.999 weight decay. During inference, we used a simple midpoint solver to solve the flow ODE, with only four steps of ODE simulation. The frequency of model inference is approximately 12Hz. The details for training and inference can be found in Table. 7

Table 7: Hyperparameters of *Flow Planner*

| Type | Parameter | Symbol | Value |
|---|---|---|---|
| Training | Num. neighboring vehicles | - | 32 |
| | Num. past timestamps | $T$ | 21 |
| | Dim. neighboring vehicles | $D_{\text{neighbor}}$ | 11 |
| | Num. lanes | - | 70 |
| | Num. points per polyline | $-$ | 20 |
| | Dim. lanes vehicles | $D_{\text{lane}}$ | 12 |
| | Num. navigation lanes | - | 25 |
| | Num. encoder block | - | 3 |
| | Num. decoder block | - | 4 |
| | Dim. encoder hidden layer | - | 192 |
| | Dim. decoder hidden layer | - | 256 |
| | Num. multi-head | - | 8 |
| | Len. trajectory segment | $L_{seg}$ | 20 |
| | Len. trajectory overlap | $L_{overlap}$ | 10 |
| Inference | Flow path | - | Conditional OT |
| | Temperature | - | 1.0 |
| | Flow ODE simulation step | - | 4 |

## D   Benchmarks and Baselines

**Benchmark Details**. For the nuPlan benchmark, we used the Val14, Test14-random and Test14-hard benchmarks for evaluation as stated in Section 4. Each of the benchmarks contains the 14 scenarios

specified by the nuPlan leaderboard respectively. The models are tested in two modes: non-reactive and reactive. In the non-reactive settings, the neighboring vehicles and other traffic participants strictly follow their trajectories in the record and will not react to the ego vehicle; on the contrary, in the reactive mode, the neighboring vehicles are controlled by the rule-based IDM [50]. However, the reaction generated by IDM can sometimes be suboptimal and lead to unnatural behavior of surrounding vehicles. For the interPlan benchmark, we chose the full-scale test set, with 335 specially augmented out-of-distribution scenarios. The models are all evaluated in reactive mode.

**Baseline Reproduction**. We followed [57] to reproduce the baselines on the nuPlan. In addition, we picked three competitive baselines for further compare on interPlan. For *Diffusion Planner*, we used the official implementation to reproduce the results, and the model is trained for 500 epochs, which is more than twice the number of *Flow Planner*.

# E    Limitation & Discussion & Future Work

The inference speed of *Flow Planner* currently represents its main limitation. While our shift from diffusion to flow matching has improved sampling efficiency, the system achieves only a speed marginally faster than 10 Hz on an A6000 GPU, where 10 Hz is the requirement for industrial deployment [16]. The main reason leading to low sampling efficiency is that the spatiotemporal self-attention mechanism remains computationally intensive. Future work will explore acceleration techniques like the shortcut model [17] to address these constraints without sacrificing planning quality. Besides, for the interactive behavior modeling, we still use an implicit design to enhance this capability. The whole method is still an imitation learning-based method, facing the problem of data quality and data forgetting issues. Maybe we can use the reinforcement learning approach to further enhance the capability of interactive behavior modeling, but there is still a long way to go to tackle problems for real-world vehicle implementation.

