# OpenReview forum: "Flow Matching-Based Autonomous Driving Planning with Advanced Interactive Behavior Modeling"
_NeurIPS.cc/2025/Conference — NeurIPS 2025 poster_

### Official Review · Reviewer_hKs2 · 2025-06-04

**Clarity:** 3
**Significance:** 3
**Originality:** 3
**Rating:** 5
**Confidence:** 4

**Summary:**

This paper introduces Flow Planner, a learning-based framework targeting the challenge of modeling interactive driving behaviors in complex scenarios. The authors identify key limitations in existing transformer-based and imitation learning approaches, particularly regarding their inability to effectively capture agent interactions due to architectural simplicity and limited interactive data. To address these gaps, the proposed method combines fine-grained trajectory tokenization, a purpose-built architecture for spatiotemporal fusion, and flow matching with classifier-free guidance to enhance multi-agent behavioral modeling. They evaluated the models on the nuPlan dataset and obtained the state-of-the-art performance.

**Questions:**

Q1. In the trajectory tokenization process, how do you ensure that each trajectory can be divided into exactly K segments with L_seg points per segment? How is this handled in cases where trajectories vary in length or have fewer points than required?

Q2. According to Table 1, several "Rule-based & Hybrid" planners achieve higher scores compared to the learning-based approach. Could you elaborate on the advantages of adopting a learning-based framework in this context, despite the relatively lower performance? What are the trade-offs in terms of generalizability, scalability, or adaptability?

Q3. The learning-based baselines in the paper are primarily focused on imitation learning. Could you clarify why optimization-based planning methods, such as reinforcement learning–based approaches, are not included for comparison? Including such baselines could provide a more comprehensive evaluation of the proposed method's effectiveness.

**Ethical Concerns:**

["NO or VERY MINOR ethics concerns only"]

**Final Justification:**

The authors' response has mainly addressed my concern. I would like to upgrade the score to 5.

**Limitations:**

yes

**Quality:**

3

**Strengths And Weaknesses:**

S1. The paper is well-organized and presented.

S2. The proposed schemes are well-examined in ablation studies.

Please check the questions part for the reviewer's confusion.

---

> ### Author Rebuttal · Authors · 2025-07-31
>
> We appreciate the constructive comments from the reviewer. In hope of addressing the reviewer's confusion, we provide the following response.
>
> > **Q1: How do you ensure that each trajectory can be divided into exactly K segments with L_seg points per segment? How is this handled in cases where trajectories vary in length or have fewer points than required?**
>
> In practical implementation, we adopt fixed trajectory lengths for both training and testing, consistent with real-world vehicle deployment. On top of this, given the predefined segment length and overlap length as hyperparameters, we can divide the trajectory into desired segments with overlap accordingly (see Section 3.2 for details). If handling variable-length inputs is required, standard padding and masking techniques can be employed to ensure consistent input dimensions across all samples.
>
> > **Q2: Could you elaborate on the advantages of adopting a learning-based framework in this context, despite the relatively lower performance? What are the trade-offs in terms of generalizability, scalability, or adaptability?**
>
> - Learning-based frameworks have largely replaced rule-based methods in academia [1] and industry [2] due to their scalability and data-driven generalization capabilities. However, trade-offs still exist: while data scales up, the scarcity of high-quality data leads to suboptimal performance in interactive scenarios and increasing computational demands when scaling up models. To address this, we propose Flow Planner—an efficient framework that achieves SOTA performance on limited nuPlan training data while maintaining fast inference speeds.
> - We further note that the **benchmark could potentially be hacked** by rule-based/hybrid methods through metric optimization - they could select trajectories specifically to maximize scores [3][4]. Flow Planner could similarly employ a post-processing module for this purpose. We implemented the same post-processing module as in [5] and achieved competitive performance with existing rule-based and hybrid methods.
>
> | Methods                     | Val14 (NR) | Val14 \(R\) | Test14-hard (NR) | Test14-hard \(R\) | Test14 (NR) | Test14 \(R\) |
> | --------------------------- | ---------- | ----------- | ---------------- | ----------------- | ---------------- | ----------------- |
> | GameFormer                  | 79.94      | 79.78       | 68.70            | 67.05             | 83.88            | 82.05             |
> | PLUTO                       | 92.88      | 76.88       | 80.08            | 76.88             | 92.23            | 90.29             |
> | Diffusion Planenr w/ refine.| 94.26      | 92.90       | 78.87            | 82.00             | 94.80            | 91.75             |
> | **Flow Planner w/ refine.**    | 94.31      | 92.38       | 78.65            | 80.25             | 94.79            | 92.4              |
>
>
> > **Q3: Could you clarify why optimization-based planning methods, such as reinforcement learning–based approaches, are not included for comparison?**
>
> As discussed in Related Works, RL has limitations such as safe exploration and reward design, which restrict its application in real world autonomous driving. In the nuPlan simulation, RL methods can leverage sufficient interactions and carefully designed rewards to achieve higher evaluation scores. However, imitation learning directly mimics the data without incorporating additional prior knowledge, making it inappropriate for direct comparison with RL approaches.
>
> > **Extra Experiment Results**
>
> To further demonstrate the effectiveness of our model in interactive behavior modeling, we conducted extra experiments on interPlan [6], a closed-loop driving benchmark extended from nuPlan, which contains 335 long-tail traffic scenarios including jaywalkers (*Jaywalk*), accidents (*Nudge Around*), and dense-lane-change situations (*High Traffic Density*) etc, **posing a huge challenge in interactive behavior modeling**. We achieved **SOTA** performance on this benchmark.
>
> | Methods                                 | Overall Score | Nudge Around | High Traffic Density | Jaywalk  |
> | --------------------------              | ----------    | ---------    | -------------------- |--------- |
> | PlanTF                                  | 47.70         | 49.40        | 58.85                | 33.94    |
> | PLUTO w/o refine.*                       | 58.47         | 71.56        | 67.25                | 25.48    |
> | Diffusion Planner                       | 52.90         | 60.48        | 49.71                | 26.20    |
> | **Flow Planner**                            | **61.82**         | 72.96        | 67.21                | 43.57    |
>
> *\*: prior knowledge is used for the model*
>
> In addition, we conducted ablation studies on interPlan to further demonstrate the improvement brought about by each key designs, proving the effectiveness of these modules in interactive behavior modeling.
>
> | Methods                                 | Overall Score | Nudge Around | High Traffic Density | Jaywalk  |
> | --------------------------              | ----------    | ---------    | -------------------- |--------- |
> | **Base Model**                              | 41.27         | 50.61        | 38.18                | 27.99    |
> | **+Fine-grained Trajectory Tokenization**   | 44.14         | 43.27        | 48.11                | 52.73    |
> | **+Scale-adaptive Attention**               | 46.25         | 51.31        | 44.28                | 34.42    |
> | **+Separate adaLN & FFN**                   | 58.22         | 64.46        | 60.08                | 43.16    |
> | **+CFG (Flow Planner)**                           | 61.82         | 72.96        | 67.21                | 43.57    |
>
>
> [1] Hu, Yihan, et al. "Planning-oriented autonomous driving." Proceedings of the IEEE/CVF conference on computer vision and pattern recognition. 2023.
>
> [2] Hwang, Jyh-Jing, et al. "Emma: End-to-end multimodal model for autonomous driving." arXiv preprint arXiv:2410.23262 (2024).
>
> [3] Cheng, Jie, Yingbing Chen, and Qifeng Chen. "Pluto: Pushing the limit of imitation learning-based planning for autonomous driving." arXiv preprint arXiv:2404.14327 (2024).
>
> [4] Dauner, Daniel, et al. "Parting with misconceptions about learning-based vehicle motion planning." Conference on Robot Learning. PMLR, 2023.
>
> [5] Zheng, Yinan, et al. "Diffusion-based planning for autonomous driving with flexible guidance." arXiv preprint arXiv:2501.15564 (2025).
>
> [6] Hallgarten, Marcel, et al. "Can vehicle motion planning generalize to realistic long-tail scenarios?." 2024 IEEE/RSJ International Conference on Intelligent Robots and Systems (IROS). IEEE, 2024.

---

### Official Review · Reviewer_bpBq · 2025-06-05

**Clarity:** 4
**Significance:** 4
**Originality:** 4
**Rating:** 6
**Confidence:** 5

**Summary:**

This paper proposes Flow Planner, a learning-based autonomous driving planner combining trajectory tokenization, spatiotemporal fusion, and classifier-free guidance. The method achieves state-of-the-art performance on nuPlan benchmarks, particularly in dense interaction scenarios like unprotected left turns. Key contributions include fine-grained trajectory decomposition and dynamic interaction reweighting during inference.

**Questions:**

N/A

**Ethical Concerns:**

["NO or VERY MINOR ethics concerns only"]

**Limitations:**

While the method shows technical soundness, its advancements over existing approaches (Diffusion Planner, UniAD) remain incremental. The lack of OOD testing and marginal benchmark improvements fail to justify the architectural complexity. However, the closed-loop interaction analysis provides value for future research directions.

**Quality:**

4

**Strengths And Weaknesses:**

Major Strength
Architectural Coherence
The integration of overlapping trajectory segments (L=20 in Table 4) provides localized interaction modeling while maintaining kinematic consistency through overlap constraints (Eq. 9). This addresses over-compression issues in prior work like Diffusion Planner.

Practical Deployment Insights
The 12Hz inference speed (Sec. 4.3) marginally exceeds industrial requirements, demonstrating awareness of real-world constraints despite computational bottlenecks.

Major Weakness
Incremental Technical Contributions
The core innovations—trajectory tokenization and scale-adaptive attention—constitute architectural refinements rather than paradigm shifts. Table 3 reveals cumulative gains of just 2.33 points from four components, suggesting diminishing returns compared to prior work, such as UniAD's 5.1-point gain over baselines.

Benchmark Limitations
Exclusive reliance on nuPlan (Table 1) raises concerns, given its known deficiencies in interaction modeling. The claimed 1.2-point improvement on Test14-hard (76.47 vs 75.22) becomes statistically marginal when considering test set variance

Minor Weaknesses
OOD Generalization Blindspot
No evaluation on navigation-focused benchmarks like Navsim, despite claims about interactive capability. This contrasts with recent works (e.g., GoalFlow'25) demonstrating 89.3 EPDMS on Navsim through explicit goal conditioning.

Safety-Critical Scenario Analysis
While Figure 2 shows collision avoidance cases, there's no quantitative breakdown of failure modes. The 27% reduction in collision rates (Table 5) lacks context regarding the remaining risk levels in production systems.

---

> ### Author Rebuttal · Authors · 2025-07-31
>
> We would like to first highlight our appreciation to the acknowledgement to our work from the reviewer. Regarding the remaining concerns of the reviewer, we provide the following response.
>
> > **W1: Architectural refinements rather than paradigm shifts**
>
> We propose a coordinated solution for interactive behavior modeling in autonomous driving, with architectural refinements as a core component. Our Fine-grained Trajectory Tokenization enables kinematically continuous trajectory representation while supporting localized feature extraction. When integrated with CFG, which dynamically reweights neighboring interactions during inference, this framework establishes a new paradigm for interactive behavior generation.
>
> > **W2, W3 & L1: Marginal improvement and Exclusive Reliance on nuPlan**
>
> - To our knowledge, Flow Planner is the first learning-based method to achieve a score above 90 on nuPlan without post-processing or prior knowledge. This accomplishment is particularly significant as the scores are averaged across 1,000+ val14 scenarios, making further improvements exceptionally challenging.
> - To further demonstrate the effectiveness of our method, we conducted additional experiments on the interPlan [1], a benchmark tailored to better evaluate the capability and robustness of algorithms **in interactive scenarios** with specially modified scenarios. Our method remained SOTA on this augmented benchmark, with a significant lead over existing methods.
>
> | Methods                                 | Overall Score | Nudge Around | High Traffic Density | Jaywalk  |
> | --------------------------              | ----------    | ---------    | -------------------- |--------- |
> | PlanTF                                  | 47.70         | 49.40        | 58.85                | 33.94    |
> | PLUTO w/o refine.*                       | 58.47         | 71.56        | 67.25                | 25.48    |
> | Diffusion Planner                       | 52.90         | 60.48        | 49.71                | 26.20    |
> | **Flow Planner**                            | **61.82**         | 72.96        | 67.21                | 43.57    |
>
> *\*: prior knowledge is used for the model*
>
> - In addition, we ablated our key designs on this benchmark to further demonstrate the effectiveness of our method in highly interactive scenarios. The proposed modules bring a total gain of **+20.55**, with a prominent improvement in interactive scenarios *Nudge Around* (**+22.35**), *High Traffic Density* (**+29.03**) and *Jaywalk* (**+15.58**) respectively. We also compared our Flow Planner with a vanilla imitation learning-based (*Vanilla IL*), which directly regresses the target trajectory, and Diffusion adaptation. Our Flow Matching-based model significantly outperforms the imitation learning (**+29.43**) and Diffusion adaptation (**+21.11**). It is worth noting that the Diffusion adaptation is trained with the same epochs of Flow Planner, which is **half** the steps of Diffusion Planner [2], indicating the advantage of fast convergence of Flow Matching.
>
> | Methods                                 | Overall Score | Nudge Around | High Traffic Density | Jaywalk  |
> | --------------------------              | ----------    | ---------    | -------------------- |--------- |
> | Vanilla IL                              | 32.39         | 50.54        | 25.19                | 32.37    |
> | Diffusion Adaptation                    | 40.71         | 25.55        | 44.21                | 40.71    |
> | **Base Model**                              | 41.27         | 50.61        | 38.18                | 27.99    |
> | **+Fine-grained Trajectory Tokenization**   | 44.14         | 43.27        | 48.11                | 52.73    |
> | **+Scale-adaptive Attention**               | 46.25         | 51.31        | 44.28                | 34.42    |
> | **+Separate adaLN & FFN**                   | 58.22         | 64.46        | 60.08                | 43.16    |
> | **+CFG (Flow Planner)**                           | 61.82         | 72.96        | 67.21                | 43.57    |
>
> - We appreciate the reviewer's suggestion to evaluate on NavSim. However, this benchmark focuses on pseudo-closed-loop testing for end-to-end driving systems, which differs from our work's scope. We would like to leave the end-to-end adaptation of our method for future study.
>
> > **W3: Inadequate Safety-Critical Scenario Analysis.**
>
> We appreciate the emphasis on safety in autonomous driving. As noted earlier, we have expanded our evaluation to include safety-critical interactive scenarios. In addition, we are confused with the reviewer's discussion on Table 5 and would like to be provided with further comments from the reviewer.
>
>
> [1] Hallgarten, Marcel, et al. "Can vehicle motion planning generalize to realistic long-tail scenarios?." 2024 IEEE/RSJ International Conference on Intelligent Robots and Systems (IROS). IEEE, 2024.
>
> [2] Zheng, Yinan, et al. "Diffusion-Based Planning for Autonomous Driving with Flexible Guidance." The Thirteenth International Conference on Learning Representations.

---

### Official Review · Reviewer_eK33 · 2025-06-18

**Clarity:** 2
**Significance:** 3
**Originality:** 3
**Rating:** 5
**Confidence:** 4

**Summary:**

The paper proposes a transformer based generative network for the planning module of an autonomous vehicle. It focuses on interactive behavior modeling by using spatiotemporal fusion from processed perception inputs and a fine-grained tokenized trajectory. In the end the model uses a classifier-free guidance by using a flow matching model to enhance interactive driving behavior by having the model predict the needed velocity to achieve two sample pairs drawn from the source and target distributions. Testing on the nuPlan dataset the model achieves SOTA results.

**Questions:**

1. How do the authors define interactive driving behavior? It is left undefined as a whole and seems quite subjective.
2. What is the multi-modality of the model? Is it just in reference to unconditioned and conditioned classifier-free guidance?

**Ethical Concerns:**

["NO or VERY MINOR ethics concerns only"]

**Final Justification:**

The authors answered the questions and addressed the issues from my review sufficiently. Thus, I keep my rating and propose the paper should be accepted.

**Limitations:**

yes

**Quality:**

3

**Strengths And Weaknesses:**

1. Quality - The paper is well written with all the claims in the paper being backed by experimental results. The experiments done are appropriate, thorough and complete. A drawback might be that the model was tested only on the nuPlan dataset and no others, but seeing that it's a standard benchmark dataset it isn't too big of a problem. Figures and graphs are legible, tables are informative.
2. Clarity - The clarity lacks a bit but it is likely due to the fact that the paper had to say so much with limited page size. The introduction leaves a bit of the proposed changes somehow undefined, while the rest of the paper answers only a portion of them.
3. Significance - The results are significant in the field in using flow matching models for generating trajectories which is a build up on diffusion models. The tokenization, spatiotemporal fusion and architectural changes to the classifier-free trajectory proposition are likely to be built upon further. The results themselves are a bit worse than, as the authors call it, rule based & hybrid models, but is the best out of the models that actually learn
4. Originality - The paper is original in the sense that it takes a base of learning trajectory planners and builds upon referenced works, while introducing modified methods for parts of the architecture. It provides new insights into trajectory planning and the way it builds upon previous work makes sense.

---

> ### Author Rebuttal · Authors · 2025-07-31
>
> We really appreciate the reviewer for the constructive comments and positive feedback on our paper.
>
> > **W1: The model was tested only on nuPlan dataset and no others.**
>
> - We thank the reviewer for the constructive comment and understanding. To provide a more comprehensive evaluation of the effectiveness of our method, we conducted extra experiments on interPlan [1], a closed-loop driving benchmark extended from nuPlan, which contains 335 long-tail traffic scenarios including jaywalkers (*Jaywalk*), accidents (*Nudge Around*), and dense-lane-change situations (*High Traffic Density*) etc, **posing a huge challenge in interactive behavior modeling**. We achieved SOTA performance on this benchmark over the existing baselines with a significant lead (**61.82**).
>
> | Methods                                 | Overall Score | Nudge Around | High Traffic Density | Jaywalk  |
> | --------------------------              | ----------    | ---------    | -------------------- |--------- |
> | PlanTF                                  | 47.70         | 49.40        | 58.85                | 33.94    |
> | PLUTO w/o refine.*                       | 58.47         | 71.56        | 67.25                | 25.48    |
> | Diffusion Planner                       | 52.90         | 60.48        | 49.71                | 26.20    |
> | **Flow Planner**                            | **61.82**         | 72.96        | 67.21                | 43.57    |
>
> *\*: prior knowledge is used for the model*
>
> - In addition, we ablated our key designs on this benchmark to further demonstrate the effectiveness of our method in highly interactive scenarios. The proposed modules bring a total gain of **+20.55**, with a prominent improvement in interactive scenarios *Nudge Around* (**+22.35**), *High Traffic Density* (**+29.03**) and *Jaywalk* (**+15.58**) respectively. We also compared our Flow Planner with a vanilla imitation learning-based (*Vanilla IL*), which directly regresses the target trajectory, and Diffusion adaptation. Our Flow Matching-based model significantly outperforms the imitation learning (**+29.43**) and Diffusion adaptation (**+21.11**). It is worth noting that the Diffusion adaptation is trained with the same epochs of Flow Planner, which is **half** the steps of Diffusion Planner [2], indicating the advantage of fast convergence of Flow Matching.
>
> | Methods                                 | Overall Score | Nudge Around | High Traffic Density | Jaywalk  |
> | --------------------------              | ----------    | ---------    | -------------------- |--------- |
> | Vanilla IL                              | 32.39         | 50.54        | 25.19                | 32.37    |
> | Diffusion Adaptation                    | 40.71         | 25.55        | 44.21                | 40.71    |
> | **Base Model**                              | 41.27         | 50.61        | 38.18                | 27.99    |
> | **+Fine-grained Trajectory Tokenization**   | 44.14         | 43.27        | 48.11                | 52.73    |
> | **+Scale-adaptive Attention**               | 46.25         | 51.31        | 44.28                | 34.42    |
> | **+Separate adaLN & FFN**                   | 58.22         | 64.46        | 60.08                | 43.16    |
> | **+CFG (Flow Planner)**                           | 61.82         | 72.96        | 67.21                | 43.57    |
>
> > **W2: Clarity issues.**
>
> We appreciate the constructive suggestion from the reviewer. We will improve the clarity of paper in the final version.
>
>
> > **Q1: How do the authors define interactive driving behavior? It is left undefined as a whole and seems quite subjective.**
>
> We appreciate the reviewer's question and apologize for any confusion caused by unclear definitions. In this work, 'interactive driving behavior' specifically refers to a vehicle's ability to perceive and respond to other road users. As previously mentioned, we conducted additional interactive scenario testing to validate the effectiveness of our design on interPlan, providing a more objective and comprehensive evaluation of our method.
>
> > **Q2: What is the multi-modality of the model? Is it just in reference to unconditioned and conditioned classifier-free guidance?**
>
> - Multi-modality driving behavior describes the diverse possible actions a driver may take in a given scenario (e.g., turning left or right to avoid an obstacle). Conventional imitation learning struggles with this as it regresses to mean ground truth values. This limitation motivates our use of probabilistic generative models like Flow Matching, whose advantage is shown in the experiment results above.
> - In terms of the Classifier-Free Guidance, it is adopted to amplify the awareness of the existence of neighboring vehicles in trajectory generation via combining the unconditioned and conditioned velocity field during inference. This novel implementation is proved to be beneficial in enhancing model's interactive modeling capability, as is shown in Table 3 and Table 5, as well as the extra experiments we provided above.
>
>
>
> [1] Hallgarten, Marcel, et al. "Can vehicle motion planning generalize to realistic long-tail scenarios?." 2024 IEEE/RSJ International Conference on Intelligent Robots and Systems (IROS). IEEE, 2024.
>
> [2] Zheng, Yinan, et al. "Diffusion-Based Planning for Autonomous Driving with Flexible Guidance." The Thirteenth International Conference on Learning Representations.

---

> > ### Comment · Reviewer_eK33 · 2025-08-04
> >
> > The authors answered the questions and addressed the issues from my review sufficiently. Thus, I keep my rating and propose the paper should be accepted.

---

> > > ### Author Response · Authors · 2025-08-04
> > > **Thanks for the acknowledgement!**
> > >
> > > Thank you for the acknowledgement to our work and the constructive comments on writing clarity and model evaluation. We will incorporate the discussion into the final version! Thanks again!

---

### Official Review · Reviewer_SPYV · 2025-06-30

**Clarity:** 3
**Significance:** 2
**Originality:** 2
**Rating:** 2
**Confidence:** 4

**Summary:**

Previous learning-based AD planning methods adopt brute-force implementation, which is not enough for  modeling interactive driving behaviors  In this work, the authors propose Flow Planner, which include coordinated designs in terms of data modeling, model architecture, and learning scheme to better model interactive driving behaviors in data. Fine-grained trajectory tokenization, temporal and spatial fusion, flow matching with classifier-free guidance are integrated into the planner and bring significant gain. Experimental results on  nuPlan dataset demonstrate the method achieves state-of-the-art performance among learning-based approaches.

**Questions:**

See the part of Weakness.

**Ethical Concerns:**

["NO or VERY MINOR ethics concerns only"]

**Final Justification:**

Thank the authors for the response.
The additional experiments on interplan are less convincing. The results show  Flow matching based model  significantly outperform diffusion based model. But as pointed out by researchers of Google DeepMind, diffusion and flow matching are similar.
And in the field of end-to-end autonous driving, diffusion-based methods achieve similar and even better  performance compared with flow matching-based method. The diffusion-based model may not be well tuned to achieve its performance upper bound.
Separate adaLN & FFN improves the score from 46.25	 to 58.22, the gain of this design is much significant than other designs. It's strange.

[1] https://diffusionflow.github.io  Diffusion Meets Flow Matching: Two Sides of the Same Coin.
[2] Diffusiondrive: Truncated diffusion model for end-to-end autonomous driving.
[3] GoalFlow: Goal-Driven Flow Matching for Multimodal Trajectories Generation in End-to-End Autonomous Driving

As for the novelty,
1. Fine-grained Trajectory Tokenization makes sense to me. But the other designs are incremental.
2. Interaction-enhanced Spatiotemporal Feature Fusion is just minor network modification.
3. Flow matching, as discussed above, is not superior to diffusion, and cannot be regarded as a contribution of this work.
4. Classifier-free guidance is well studied. Conditioning on neighboring vehicles is an incremental design.
Overall, I still hold the concern about novelty and 'will keep the original rating.

**Limitations:**

1. This work is like a combination of several incremental designs, The novelty  is limited. It's the main concern.
2. The experiment parts are not convincing enough.

**Paper Formatting Concerns:**

I have no paper formatting concerns.

**Quality:**

2

**Strengths And Weaknesses:**

Strength:
1. The paper is well written and easy to follow.
2. Impressive and intuitive figures to show the detailed designs.

Weakness:
1.  This work is like a combination of several incremental designs, like flow matching mechanism, trajectory tokenization manner, and classifier-free guidance,  and feature fusion manner.  And most of the designs are already well studied and proved to be effective in other works, especially flow matching and classifier-free guidance, which cannot be regarded as contributions of this work. Overall, the novelty of this work is limited and incremental. It's the main concern.
2. The experiments are not adequate to show the effectiveness of the paper, only on one dataset with comparisons with few baseline methods. And the ablations in Table 3 show that many of the proposed components have marginal improvements.
3. The layouts of table 4 and figure 3 can be improved.

---

> ### Author Rebuttal · Authors · 2025-07-31
>
> We thank the reviewer for the constructive comments. Regarding the concerns of the reviewer, we provide the following responses with extra experiments.
>
> To further demonstrate the effectiveness of our design, we used interPlan [1], a closed-loop driving benchmark extended from nuPlan, which contains 335 long-tail traffic scenarios including jaywalkers (*Jaywalk*), accidents (*Nudge Around*), and dense-lane-change situations (*High Traffic Density*) etc, **posing a huge challenge in interactive behavior modeling**. We remain **SOTA** on this benchmark, with a significant lead over previous methods.
>
> | Methods                                 | Overall Score | Nudge Around | High Traffic Density | Jaywalk  |
> | --------------------------              | ----------    | ---------    | -------------------- |--------- |
> | PlanTF                                  | 47.70         | 49.40        | 58.85                | 33.94    |
> | PLUTO w/o refine.*                       | 58.47         | 71.56        | 67.25                | 25.48    |
> | Diffusion Planner                       | 52.90         | 60.48        | 49.71                | 26.20    |
> | **Flow Planner**                            | **61.82**         | 72.96        | 67.21                | 43.57    |
>
> *\*: prior knowledge is used for the model*
>
> In addition, we conducted further ablation studies on the interPlan benchmark, and compared our model with a vanilla imitation learning-based model *Vanilla IL*, which directly regress the target trajectory, and a Diffusion adaptation. **The proposed modules bring significant performance improvement in interactive scenairos**, and our method outperforms the imitation learning-based and diffusion adaptation with a prominent margin.
>
> | Methods                                 | Overall Score | Nudge Around | High Traffic Density | Jaywalk  |
> | --------------------------              | ----------    | ---------    | -------------------- |--------- |
> | Vanilla IL                              | 32.39         | 50.54        | 25.19                | 32.37    |
> | Diffusion Adaptation                    | 40.71         | 25.55        | 44.21                | 40.71    |
> | **Base Model**                              | 41.27         | 50.61        | 38.18                | 27.99    |
> | **+Fine-grained Trajectory Tokenization**   | 44.14         | 43.27        | 48.11                | 52.73    |
> | **+Scale-adaptive Attention**               | 46.25         | 51.31        | 44.28                | 34.42    |
> | **+Separate adaLN & FFN**                   | 58.22         | 64.46        | 60.08                | 43.16    |
> | **+CFG (Flow Planner)**                           | 61.82         | 72.96        | 67.21                | 43.57    |
>
> > **W1 & L1: The novelty of this work is limited and incremental.**
>
> Interactive behavior modeling remains a fundamental challenge in autonomous driving. Our core contribution is an integrated framework where each component is carefully designed to enhance interactive behavior modeling.
>
> <!-- It is unfair to claim there is a lack of novelty. -->
>
> - `Fine-grained Trajectory Tokenization` Our Fine-grained Trajectory Tokenization (acknowledged by reviewer yVKT & bpBq) enables kinematically continuous trajectory representation while supporting localized feature extraction, providing **a solution to the overcompression problem in existing methods [2][3][4]**. The Fine-grained Trajectory Tokenization enables high-quality and flexible trajectory generation (Fig. 3) and **brings performance improvement in interactive scenarios**, as is indicated in the table above.
> - `Interaction-enhanced Spatiotemporal Feature Fusion` Our Interaction-enhanced Spatiotemporal Feature Fusion (acknowledged by reviewer eK33) effectively integrates heterogeneous scene information through cooperative design with trajectory tokenization, **contrasting with brute-force transformer blocks[5] or unnecessarily complex architectures in prior work[6]**. This mechanism prominently enhances model's robustness, with a **+14.08** gain in overall performance on interPlan.
> - `Flow matching` Flow Matching-based generative models excel at capturing multi-modal driving behavior, essential for modeling complex interactive scenarios in autonomous driving[2][3]. Their straightforward loss function leads to faster convergence, more stable training, and higher inference efficiency than diffusion models [7]. As shown in the table, our Flow Matching model outperforms both the vanilla imitation learning model (**+29.43**) and the Diffusion adaptation (**+21.11**), **demonstrating the effectiveness of Flow Matching-based model in modeling interactive behavior**. Notably, the Diffusion adaptation is trained for the same 200 epochs as Flow Planner—only **half** the steps used by Diffusion Planner [2]—highlighting Flow Matching’s training efficiency.
> - `Classifier-free guidance` While CFG is widely used in diffusion models, our work **novelly conditions on neighboring vehicles**, effectively enhancing interactive behavior modeling for autonomous driving (acknowledged by reviewer bpBq & eK33). Unlike typical CFG use in tasks with explicit conditions (e.g., text prompts [8] or rewards [9]), **interaction in driving lacks clear definitions**. We find that neighboring vehicles provide a powerful implicit condition. By learning both unconditioned and scene-conditioned behaviors, CFG captures interaction patterns—**offering a novel solution for generating interactive behaviors**. As shown, our model achieves **61.82** on interPlan vs. 58.22 without CFG, with notable gains in *Nudge Around* (**+8.50**) and *High Traffic Density* (**+7.13**).
>
> In hope of better addressing the reviewer's concerns, we welcome any further, more specific questions or comments regarding the novelty of our work.
>
> > **W2 & L2: The experiments are not adequate to show the effectiveness of the paper.**
>
> - We evaluate our method on nuPlan, the largest open-source real-world driving dataset supporting closed-loop testing. As the standard benchmark for planning methods, nuPlan better aligns with real-world applications. Our comparisons already have included existing SOTA methods.
> - To demonstrate the effectiveness of our designs, we further compared our method with existing competitive models on interPlan. We remain SOTA performance on this benchmark, with a significant lead (**61.82**) over existing models, demonstrating the effectiveness and robustness **in interactive behavior modeling** of our designs. The additional ablation on interPlan further indicates the effectiveness in interactive scenarios of each module we propose, as shown in the table above and discussion on W1 & L1.
>
> > **W3: The layouts of table 4 and figure 3 can be improved**
>
> We appreciate the constructive suggestion from the reviewer. We will improve the layouts of the mentioned table and figure in the final version of our paper.
>
> [1] Hallgarten, Marcel, et al. "Can vehicle motion planning generalize to realistic long-tail scenarios?." 2024 IEEE/RSJ International Conference on Intelligent Robots and Systems (IROS). IEEE, 2024.
>
> [2] Zheng, Yinan, et al. "Diffusion-Based Planning for Autonomous Driving with Flexible Guidance." The Thirteenth International Conference on Learning Representations.
>
> [3] Liao, Bencheng, et al. "Diffusiondrive: Truncated diffusion model for end-to-end autonomous driving." Proceedings of the Computer Vision and Pattern Recognition Conference. 2025.
>
> [4] Cheng, Jie, Yingbing Chen, and Qifeng Chen. "Pluto: Pushing the limit of imitation learning-based planning for autonomous driving." arXiv preprint arXiv:2404.14327 (2024).
>
> [5] Cheng, Jie, et al. "Rethinking imitation-based planners for autonomous driving." 2024 IEEE International Conference on Robotics and Automation (ICRA). IEEE, 2024.
>
> [6] Huang, Zhiyu, Haochen Liu, and Chen Lv. "Gameformer: Game-theoretic modeling and learning of transformer-based interactive prediction and planning for autonomous driving." Proceedings of the IEEE/CVF International Conference on Computer Vision. 2023.
>
> [7] Lipman, Yaron, et al. "Flow matching for generative modeling." arXiv preprint arXiv:2210.02747 (2022).
>
> [8] Ramesh, Aditya, et al. "Hierarchical text-conditional image generation with clip latents." arXiv preprint arXiv:2204.06125 1.2 (2022): 3.
>
> [9] Ajay, Anurag, et al. "Is conditional generative modeling all you need for decision-making?." arXiv preprint arXiv:2211.15657 (2022).

---

> > ### Comment · Reviewer_SPYV · 2025-08-05
> >
> > Thank the authors for the response.
> > The additional experiments on interplan are less convincing. The results show  Flow matching based model  significantly outperform diffusion based model. But as pointed out by researchers of Google DeepMind, diffusion and flow matching are similar.
> > And in the field of end-to-end autonous driving, diffusion-based methods achieve similar and even better  performance compared with flow matching-based method. The diffusion-based model may not be well tuned to achieve its performance upper bound.
> > Separate adaLN & FFN improves the score from 46.25	 to 58.22, the gain of this design is much significant than other designs. It's strange.
> >
> > [1] https://diffusionflow.github.io  Diffusion Meets Flow Matching: Two Sides of the Same Coin.
> > [2] Diffusiondrive: Truncated diffusion model for end-to-end autonomous driving.
> > [3] GoalFlow: Goal-Driven Flow Matching for Multimodal Trajectories Generation in End-to-End Autonomous Driving
> >
> > As for the novelty,
> > 1. Fine-grained Trajectory Tokenization makes sense to me. But the other designs are incremental.
> > 2. Interaction-enhanced Spatiotemporal Feature Fusion is just minor network modification.
> > 3. Flow matching, as discussed above, is not superior to diffusion, and cannot be regarded as a contribution of this work.
> > 4. Classifier-free guidance is well studied. Conditioning on neighboring vehicles is an incremental design.
> > Overall, I still hold the concern about novelty and 'will keep the original rating.

---

> > > ### Author Response · Authors · 2025-08-06
> > > **Response to Reviewer SPYV**
> > >
> > > Thank you for the further comment, and below we provide the following response respectively:
> > >
> > > 1. We do not intend to highlight the superiority of performance of Flow Matching and Diffusion in our work. Instead, we would like to reiterate that we adopt the Flow Matching loss for its advantage in more stable training, faster convergence and higher sample efficiency [7] (in the rebuttal) [2], compared with Diffusion models.
> > > 2. On top of this, we trained a Diffusion adaptation of our model using the exact same settings (e.g. same training steps) with Flow Planner, and the results can further indicate the advantage of Flow Matching in fast convergence, which achieves better performance with the same training steps.
> > > 3. In terms of the comparison of [2] and [3] (by the reviewer), these approaches exhibit additional differences in their structural choices and design principles beyond the application of Diffusion and Flow Matching (e.g [2] uses "truncated Diffusion" with predefined anchors), and directly comparing these two methods fails to isolate the benefits brought specifically by Diffusion and Flow Matching. We would also like to point out that Flow Matching-based [3] actually outperforms [2] on end-to-end benchmark, demonstrating the strength of Flow Matching.
> > > 4. The ablation is performed by adding key components one at a time, and the introduction of Separate adaLN and FFN leads to enhancements in structural design as well as an increase in model size. The combination of these two factors results in substantial performance gains.
> > > 5. We respect the reviewer's opinion on the contribution of our work. However, we hope the reviewer can provide constructive comments for more productive discussion.
> > >
> > > [1] Lipman, Yaron, et al. "Flow matching guide and code." arXiv preprint arXiv:2412.06264 (2024)
> > >
> > > [2] Liu, Xingchao, Chengyue Gong, and Qiang Liu. "Flow straight and fast: Learning to generate and transfer data with rectified flow." arXiv preprint arXiv:2209.03003 (2022).

---

### Official Review · Reviewer_yVKT · 2025-07-02

**Clarity:** 3
**Significance:** 3
**Originality:** 2
**Rating:** 3
**Confidence:** 4

**Summary:**

This paper proposes structural improvements across trajectory representation, heterogeneous context fusion, and training
methodology to enhance interactive behavior modeling in autonomous driving planners. First, the authors introduced a Fine-grained Trajectory Tokenization method, which segments the ego vehicleʼs future trajectory into multiple tokens. This design allows for a more precise representation of spatiotemporal interactions. Second, the paper presents an Interaction-enhanced Spatiotemporal Fusion Block that accounts for the statistical heterogeneity across different modalities such as lanes, neighboring agents, and the ego vehicle itself. This block incorporates modality-specific normalization (adaLN, distance-aware attention mechanism (scale-adaptive attention), and separate feed forward networks for each modality, enabling more efficient and focused information fusion. Lastly, to improve training efficiency, the
authors adopt a Flow Matching framework and apply Classifier-Free Guidance (CFG) at inference time. By jointly training conditioned and
unconditioned outputs within a single model, the approach enables adjustment of interaction sensitivity during inference. The proposed
model achieves a state-of-the-art CLS-NR score of 90.4 on the nuPlan Val14 benchmark.

**Questions:**

Please refer to the weakness part above. Authors could respond to my concerns.

**Ethical Concerns:**

["NO or VERY MINOR ethics concerns only"]

**Final Justification:**

It remains unclear whether the proposed methods effectively achieve the goal of improved interaction modeling. Therefore, I will maintain my current rating.

**Limitations:**

yes.

**Quality:**

3

**Strengths And Weaknesses:**

Strength
- This paper structurally adapts and integrates several techniques originally developed for diffusion-based generative models—such as
Flow Matching, Classifier-Free Guidance (CFG), and adaptive Layer Normalization (adaLN) to the task of autonomous driving planning.  These methods are incorporated effectively within the proposed framework.
- The paper introduces a novel Fine-grained Trajectory Tokenization approach that segments the ego vehicleʼs future trajectory into multiple
temporally ordered segments, each treated as an individual token. This enables the Transformerʼs attention mechanism to capture
spatiotemporal interactions more precisely. The proposed design is structurally meaningful and brings novelty in improving interactive behavior modeling.

Weakness
- Limited Novelty and Unclear Effectiveness of the Interaction-enhanced Spatiotemporal Fusion Block:  While the proposed Fusion Block introduces scale-adaptive attention to the planning task for the first time, the method itself has already been explored in other domains and thus lacks structural novelty.
- Moreover, computing attention scores solely based on pairwise distances between entities may not sufficiently capture qualitative relationships or context-aware complex interactions between agents.  As a result, the contribution of this module to improving interactive
behavior modeling appears limited, and the structural design does not offer a novel solution.
- Limited Relevance of Flow Matching & Classifier-Free Guidance to Interactive Behavior Modeling: Flow Matching, originally proposed to improve training efficiency in diffusion-based models, is applied here to the planning task. However, it does not directly address or improve interactive behavior modeling in a structural sense. CFG is used during inference to blend conditioned and unconditioned outputs, amplifying the influence of  scene-level conditioning. While this technique is effective in some planning scenarios, it may not a fundamentally novel or targeted solution for modeling interactive behavior.

---

> ### Author Rebuttal · Authors · 2025-07-31
>
> We thank the reviewer for the constructive comments. To further address the concerns of reviewer, we conducted extra experiments and provide the following response.
>
> We used interPlan [1], a closed-loop driving benchmark extended from nuPlan, which contains 335 long-tail traffic scenarios including jaywalkers (*Jaywalk*), accidents (*Nudge Around*), and dense-lane-change situations (*High Traffic Density*) etc, **posing a huge challenge in interactive behavior modeling**. **We achieved SOTA performance on the interPlan benchmark**.
>
>
> | Methods                                 | Overall Score | Nudge Around | High Traffic Density | Jaywalk  |
> | --------------------------              | ----------    | ---------    | -------------------- |--------- |
> | PlanTF                                  | 47.70         | 49.40        | 58.85                | 33.94    |
> | PLUTO w/o refine.*                      | 58.47         | 71.56        | 67.25                | 25.48    |
> | Diffusion Planner                       | 52.90         | 60.48        | 49.71                | 26.20    |
> | **Flow Planner**                            | **61.82**         | 72.96        | 67.21                | 43.57    |
>
> *\*: prior knowledge is used for the model*
>
> We conducted further ablation studies on the interPlan by adding the modules onto the base model. We also compared our model with an vanilla imitation learning-based model *Vanilla IL*, which shares the same model architecture but directly regresses the trajectory, and a Diffusion adaption of our model. **The proposed modules bring significant performance improvement**, and the result demonstrates the superiority of Flow Matching-based paradigm compared with vanilla imitation learning and Diffusion for interactive behavior modeling.
>
> | Methods                                 | Overall Score | Nudge Around | High Traffic Density | Jaywalk  |
> | --------------------------              | ----------    | ---------    | -------------------- |--------- |
> | Vanilla IL                              | 32.39         | 50.54        | 25.19                | 32.37    |
> | Diffusion Adaptation                    | 40.71         | 25.55        | 44.21                | 40.71    |
> | **Base Model**                              | 41.27         | 50.61        | 38.18                | 27.99    |
> | **+Fine-grained Trajectory Tokenization**   | 44.14         | 43.27        | 48.11                | 52.73    |
> | **+Scale-adaptive Attention**               | 46.25         | 51.31        | 44.28                | 34.42    |
> | **+Separate adaLN & FFN**                   | 58.22         | 64.46        | 60.08                | 43.16    |
> | **+CFG (Flow Planner)**                           | 61.82         | 72.96        | 67.21                | 43.57    |
>
> > **W1 & W2: Limited Novelty and Unclear Effectiveness of the Interaction-enhanced Spatiotemporal Fusion Block and Scale-adaptive Attention**
>
> We propose a **coordinated solution** for interactive behavior modeling in autonomous driving, with our Interaction-enhanced Spatiotemporal Fusion Mechanism serving as a key component.
> - As a subcomponent, **our architecture is not naively implemented**. Crucially, it coordinates with Fine-grained Trajectory Tokenization to enable richer bidirectional interactions among heterogeneous tokens. Besides, with the help of Scale-adaptive Attention, traffic participants within a certain distance are assigned higher attention scores, enabling each trajectory token to capture spatiotemporal interactions more precisely and adaptively. All these design enhances the expressiveness of interactive behavior modeling while serving as the foundation for flow matching and classifier-free guidance designs.
> - Previous methods attempt to enhance interactive behavior modeling by employing complex game-theoretic models [2] or topology-based representations [3] to capture intricate interactive behaviors. In contrast, **we pursue an elegant yet effective architecture** that demonstrates its effectiveness through our method's SOTA performance (90.43), **achieving marked improvement** over vanilla Transformer approaches (PlanTF [4], 84.27) and complex specialized architectures (GameFormer [2], 13.32). Our model also achieves SOTA performance (61.82) on interPlan benchmark with the significant performance gain (**+14.08**) brought about by the two key designs.
>
> > **W3: Limited Relevance of Flow Matching & Classifier-Free Guidance (CFG) to Interactive Behavior Modeling**
>
>
>
> - `W3.1 Limited Relevance of Flow Matching` Generative models demonstrate superior multimodal behavior modeling capabilities, which are fundamental for interactive behavior modeling [5][6]. On top of this, Flow Matching provides more stable training and faster convergence, as noted by the reviewer. As is shown in the table above, when incorporated with our architecture, our method (**61.82**) outperforms the imitation-learning based model (32.39) and the diffusion adaptation (40.71) with a significant lead **in interactive scenarios**. Note that the Diffusion adaptation is trained with the same epochs (200 epochs) with Flow Planner, only **half** the steps of Diffusion Planner [5], indicating the advantage of faster convergence of Flow Matching.
> - `W3.2 Limited Relevance of CFG` While CFG is widely used in diffusion models, our work **novelly conditions on neighboring vehicles**, effectively enhancing interactive behavior modeling for autonomous driving. Unlike typical CFG use in tasks with explicit conditions (e.g., text prompts [7] or rewards [8]), **interaction in driving lacks clear definitions**. We find that neighboring vehicles provide a powerful implicit condition. By learning both unconditioned and scene-conditioned behaviors, CFG captures interaction patterns—**offering a novel solution for generating interactive behaviors**. As shown, our model achieves **61.82** on interPlan vs. 58.22 without CFG, with notable gains in *Nudge Around* (**+8.50**) and *High Traffic Density* (**+7.13**).
>
>
>
> [1] Hallgarten, Marcel, et al. "Can vehicle motion planning generalize to realistic long-tail scenarios?." 2024 IEEE/RSJ International Conference on Intelligent Robots and Systems (IROS). IEEE, 2024.
>
> [2] Huang, Zhiyu, Haochen Liu, and Chen Lv. "Gameformer: Game-theoretic modeling and learning of transformer-based interactive prediction and planning for autonomous driving." Proceedings of the IEEE/CVF International Conference on Computer Vision. 2023.
>
> [3] Liu, Haochen, et al. "Reasoning multi-agent behavioral topology for interactive autonomous driving." Advances in Neural Information Processing Systems 37 (2024): 92605-92637.
>
> [4] Cheng, Jie, et al. "Rethinking imitation-based planners for autonomous driving." 2024 IEEE International Conference on Robotics and Automation (ICRA). IEEE, 2024.
>
> [5] Zheng, Yinan, et al. "Diffusion-Based Planning for Autonomous Driving with Flexible Guidance." The Thirteenth International Conference on Learning Representations.
>
> [6] Liao, Bencheng, et al. "Diffusiondrive: Truncated diffusion model for end-to-end autonomous driving." Proceedings of the Computer Vision and Pattern Recognition Conference. 2025.
>
> [7] Ramesh, Aditya, et al. "Hierarchical text-conditional image generation with clip latents." arXiv preprint arXiv:2204.06125 1.2 (2022): 3.
>
> [8] Ajay, Anurag, et al. "Is conditional generative modeling all you need for decision-making?." arXiv preprint arXiv:2211.15657 (2022).

---

> > ### Comment · Reviewer_yVKT · 2025-08-07
> > **Further thought**
> >
> > Thank you for providing responses to my comments.
> > However, it still remains unclear how interaction modeling is specifically improved in this work. The primary novel contribution appears to be trajectory tokenization, but it is not evident how the use of overlapped trajectory tokens contributes to enhanced interaction modeling. Furthermore, the rationale behind employing flow matching with classifier-free guidance over traditional diffusion networks lacks sufficient justification. A clearer explanation of the design choices and a more in-depth behavior analysis are necessary to position the proposed approach within the current landscape.

---

### Decision · Program_Chairs · 2025-09-17

**Decision:**

Accept (poster)

**Comment:**

The paper has received a mixture of opposite reviews ranging from reject to strong accept. All reviewers acknowledge the clarity of the paper, and the benefits of the results. There is, however, a incremental novelty concern shared among different reviewers. Nevertheless, overall feels like a solid paper.